# Regulating interfacial reaction through electrolyte chemistry enables gradient interphase for low-temperature zinc metal batteries

Wei Wang[1], Shan Chen[1], Xuelong Liao[1], Rong Huang[1], Fengmei Wang[2], Jialei Chen[1], Yaxin Wang[1], Fei Wang [2] ✉ & Huan Wang [1] ✉

In situ formation of a stable interphase layer on zinc surface is an effective solution to suppress dendrite growth. However, the fast transport of bivalent Zn-ions within the solid interlayer remains very challenging. Herein, we engineer the SEI components and enable superior kinetics of Zn metal batteries under harsh conditions through regulating the sequence of interfacial chemical reaction. With the differences in chemical reactivity of trimethyl phosphate co-solvent and trifluoromethanesulfonate anions in the $Zn^{2+}$-solvation shell, $Zn_3(PO_4)_2$ and $ZnF_2$ are successively generated on Zn metal surface to form a gradient $ZnF_2–Zn_3(PO_4)_2$ interphase. Mechanistic studies reveal the outer $ZnF_2$ facilitates $Zn^{2+}$ desolvation and inner $Zn_3(PO_4)_2$ serves as channels for fast $Zn^{2+}$ transport, contributing to long-term cycling at subzero temperatures. Impressively, the gradient SEI enables a high lifespan over 7000 hours in Zn symmetric cell and a capacity retention of 86.1% after 12000 cycles in Zn–KVOH full cell at −50 °C.

The ever-increasing demand for renewable energy sources (such as solar and wind) and their fluctuating nature necessitate the development of grid-scale energy storage technologies to minimize fossil fuel consumption. Aqueous zinc (Zn) metal batteries (ZMBs) have been prospected as appealing choices for enabling more economical and large-scale battery systems due to their abundant reserves, low manufacturing cost, high safety and intrinsic merits of Zn metal, including high theoretical capacity ($820\,mAh\,g^{-1}$) and low redox potential (−0.76 V versus the standard hydrogen electrode)[1–5]. However, in terms of stationary energy storage applications in cold climates or high-latitude regions rich in renewable energy, the ZMBs suffer from severe performance degradation and even fail to work due to the solidification of the aqueous electrolyte and deteriorated electrode/electrolyte interphase at subzero temperatures[6–9].

The hydrogen (H)-bonds between water molecules become stronger as temperature drops, which drives the disordered water into ordered ice[10,11]. Thus, tuning the electrolyte compositions to break the intermolecular H-bonds in water has been regarded as a general principle to prevent water from solidifying. In this respect, the prevailing research focuses on the following three approaches: (i) employing concentrated aqueous electrolyte that can provide abundant ions to bond with O−H in water[7,12–14]; (ii) introducing additives/cosolvents to reform H-bonds with water[8,15–18]; (iii) fabricating hydrogel to intensify the interaction with water[19,20]. These strategies have enabled the water-based electrolyte with good low-temperature adaptability, but the cycling life of low-temperature Zn metal anode is generally limited to hundreds of hours and the limited utilization of Zn, which is far from satisfactory for practical implementation.

[1]Key Laboratory of Advanced Energy Materials Chemistry (Ministry of Education), Renewable Energy Conversion and Storage Center (RECAST), College of Chemistry, Nankai University, 300071 Tianjin, China. [2]Department of Materials Science, Fudan University, 200433 Shanghai, China. ✉e-mail: feiw@fudan.edu.cn; huan.wang0520@nankai.edu.cn

The chemically instable nature of Zn metal against water causes the continuous and slow $H_2$ reaction at low temperatures. This can result in the local enhancement of $OH^-$ concentration, which in turn corrodes the Zn metal surface and forms a loose and plate-like passivation film, reducing $Zn^{2+}$ transport kinetics and leading to uncontrolled Zn dendrite growth[21–23]. In this scenario, building a dense solid electrolyte interphase (SEI) that can block water penetration is the most effective route to suppress the parasitic reactions and ensure durable Zn metal anodes[24,25]. However, learning from the former experiences on low-temperature lithium (Li) batteries[26–28], the interface kinetics associated with $Zn^{2+}$ desolvation and conduction are extremely sluggish at low temperatures, which are mainly responsible for the poor cycling performance. Compared with Li-ions, it is more challenging for the bivalent Zn-ions to cross the as-formed SEI at subzero temperatures, thus resulting in huge cell impedance, non-uniform deposition and severe capacity decay. Therefore, regulating the components and distribution of the as-formed SEI on the Zn surface to lower the energy barrier of $Zn^{2+}$ desolvation and its transport through the SEI is the key premise for achieving long-term stable cycling behaviors under low-temperature condition, however, has rarely been explored.

Engineering electrolyte chemistry can simultaneously mitigate the two issues of low-temperature ZMBs, including water solidification and SEI-related kinetics, which is also the simplest approach that could be easily adapted to practical applications. With the merits of high Gutmann donor number, fire retardance and low viscosity, trimethyl phosphate (TMP) has been selected as the primary organic solvent or cosolvent into the aqueous electrolyte to form $Zn_3(PO_4)_2$ for dendrite-free ZMBs[29,30]. However, it is rather difficult for single $Zn_3(PO_4)_2$ to simultaneously possess low $Zn^{2+}$-desolvation energy and conduction barrier towards fast $Zn^{2+}$ kinetics. On the other hand, the anions of salt can also coordinate with $Zn^{2+}$ and contribute to the SEI formation. Yet, how to tailor the difference in reduction potential of cosolvent and coordinated anions for kinetically favorable SEI remains elusive.

Herein, we employ TMP as a cosolvent into the aqueous electrolyte comprising 2 M of zinc trifluoromethanesulfonate ($Zn(OTf)_2$) for anti-freezing and long-life ZMBs. The TMP co-solvent can not only break the H-bonds to endow the hybrid electrolyte with a low freezing point of −56.8 °C but also can regulate the $Zn^{2+}$-solvation structure with a configuration of $Zn^{2+}[H_2O]_{5.02}[TMP]_{0.14}[OTf^-]_{0.84}$. Moreover, the lowest unoccupied molecular orbital (LUMO) energy of $Zn^{2+}$–TMP in the solvation shell is lower than that of $Zn^{2+}$–$OTf^-$, which can be preferentially reduced on the Zn metal surface, followed by the reductive reaction of $OTf^-$, thus forming the gradient interphase with $Zn_3(PO_4)_2$ in the bottom and $ZnF_2$ on the top. Both the experimental characterizations and calculation results reveal that the outer $ZnF_2$ promotes the desolvation of $Zn^{2+}$ on the interface and inner $Zn_3(PO_4)_2$ facilitates rapid transport across the SEI, respectively. Besides, it is found that the subzero temperature is beneficial for the formation of more uniform and denser SEI due to the considerable suppression of water-associated side reactions. As a result of the joint effects, it achieves an average Coulombic efficiency of 99.9% over 3800 cycles and a high Zn utilization rate of 94% at −30 °C, and remarkable durability over 7000 h at −50 °C, which represent the best low-temperature ZMBs performance to the best of our knowledge. Furthermore, high-capacity full cells with KVOH and $MnO_2$ as cathodes were also demonstrated with superb capacity retention ability.

## Results

### Exploring optimal formulation for low-temperature aqueous electrolyte

A series of TMP/water hybrid electrolytes with 2 M $Zn(OTf)_2$ were prepared, where the volume percentage of TMP ranges from 0%, 5%, 10%, 20%, 40%, 60%, 80% to 100%, and the corresponding electrolyte is marked as TMP–0, TMP–5, TMP–10, TMP–20, TMP–40, TMP–60,

TMP–80 and TMP–100, respectively. After 3 h of resting at −50 °C, the TMP–40, TMP–60 and TMP–80 can maintain the liquid state without deposit or phase separation, whereas crystallization/solidification was observed in the other counterparts (Fig. 1a). Further, the differential scanning calorimeter (DSC) curves show the freezing point of the hybrid electrolyte is lowered to −56.8 °C as the TMP content reaches 40% (Fig. 1b and Supplementary Fig. 1), which can be explained by the breakage of H-bonds network in water molecules by TMP co-solvent. On the other hand, the ionic conductivity negatively correlates to the content of TMP in temperatures ranging from −30 °C to 60 °C (Fig. 1c, Supplementary Fig. S2, and Supplementary Table S1), possibly due to a slight increase in electrolyte viscosity. However, the TMP–0 exhibits a sudden drop in ionic conductivity below −30 °C, because of the high freezing point (−34.8 °C in DSC) that causes the electrolyte solidification. In sharp contrast, the TMP–40 exhibits a slow decline in ionic conductivity as the temperature decreases and achieves a high ionic conductivity of 0.85 mS cm$^{-1}$ even at −50 °C (>0.1 mS cm$^{-1}$)[28], several orders of magnitude higher than that of TMP–0.

To explore the molecular interaction within the hybrid electrolytes, a series of spectroscopic characterizations were performed. The Raman peaks associated with O–H stretching vibration of water molecules are visible within the range from 3100 to 3800 cm$^{-1}$, which can be divided into three peaks, including strong, weak and non-H-bonds (Fig. 1d, e and Supplementary Figs. S3 and S4)[7,31]. It was found that the probability of non-H bonds increases with TMP content, while the change of strong-H bonds shows the opposite trend (Fig. 1f). Notably, as the TMP content was increased to 40%, the percentage of non-H-bond almost reached the maximum. Moreover, the Fourier transform infrared (FTIR) results (Supplementary Fig. S5) show that the O–H and C–H stretching vibration modes experience significant blueshifts and redshifts with the increase of TMP, respectively, largely ascribed to the breakage of H-bond in water along with the H-bond formation between TMP and water in the hybrid electrolyte[8]. This can be further confirmed by the $^1H$ nuclear magnetic resonance (NMR) spectra (Fig. 1g and Supplementary Fig. S6), wherein the $^1H$ from $H_2O$ and TMP both chemical shift to a low field with the increase of TMP. These results reveal that the TMP can interact with water to reform H-bonds, during which the H-bonds in water were largely destroyed, thus affording a low freezing point and high ionic conductivity of the hybrid electrolyte at low temperatures. To maximally inherit the unique merits of aqueous electrolyte, TMP–40 is considered the optimal electrolyte formulation for low-temperature ZMBs. Also, the TMP–40 endows the separator with high fire retardance (Supplementary Fig. S7), indicating the hybrid electrolyte is safe enough to operate.

### Solvation structure and SEI characterization

Since the SEI components highly depend on the solvation sheath of $Zn^{2+}$, we furthered the study of $Zn^{2+}$ solvation structure in a series of TMP/$H_2O$ electrolytes through theoretical calculations and experimental characterizations. The Raman characterizations show the $SO_3$ stretching band in the $OTf^-$–anions experiences a gradual shift with the increase of TMP concentration (Fig. 2a and Supplementary Fig. S8). The broad peaks can be well fitted into three peaks at ~1028 cm$^{-1}$, ~1033 cm$^{-1}$, and ~1040 cm$^{-1}$, corresponding to the free anion (FA, $OTf^-$), solvent-separated ion pairs (SSIP, $Zn^{2+}$–$(H_2O)_x(TMP)_y$–$OTf^-$) and contact ion pairs (CIP, $Zn^{2+}$–$OTf^-$), respectively[32,33], as shown in Supplementary Fig. S9. By calculating the peak area ratio, the CIP percentage increases with the increase of TMP concentration and reaches the maximum value of 51.92% with 40% of TMP (Supplementary Fig. S10), indicating $OTf^-$ anion is involved in the $Zn^{2+}$ solvation sheath. Afterwards, the CIP content decreases with the increase of TMP, possibly due to the strong binding of TMP and $Zn^{2+}$ that causes more TMP to enter the $Zn^{2+}$ solvation sheath by substituting partial $OTf^-$ anions. Meanwhile, there is a V-shape relationship between FA percentage and TMP concentration, where the lowest FA ratio is 6.59%, suggestive of

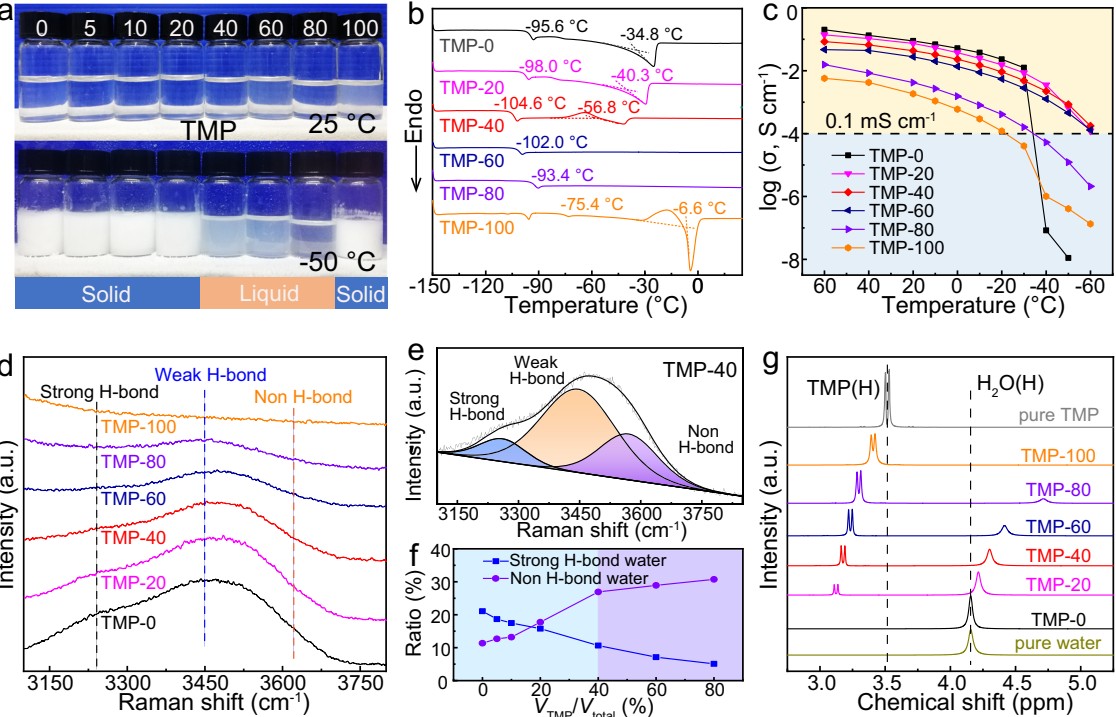

**Fig. 1 | Characterizations of the TMP/water hybrid electrolytes with different TMP contents to explore the optimal formulation for low-temperature ZMBs.** **a** The photographs of hybrid electrolytes at 25 °C (top) and −50 °C (bottom). **b** DSC curves to obtain the freezing points. **c** Ionic conductivities at different temperatures. **d** The Raman spectra of O–H stretching vibration. **e** The fitted O–H stretching vibration of TMP–40 electrolyte. **f** Ratios of strong H-bond and non-H-bond water with TMP content in the hybrid electrolyte. **g** $^1H$ chemical shift of water with different TMP contents.

more OTf⁻ involved in the solvated structure of $Zn^{2+}$. This also confirms that TMP–40 is the optimized electrolyte formulation for the in situ formation of favorable SEI to suppress side reactions and facilitate $Zn^{2+}$ transport. For the Raman spectra of the TMP, the P–O–(C) symmetric stretching vibration gradually blueshifts as the increase of TMP (Fig. 2b and Supplementary Fig. S11), indicating more TMP participates in the $Zn^{2+}$ solvation shell[34,35], according well with the above results. This is also supported by the higher binding energy of $Zn^{2+}$–TMP complex (−200.36 kJ mol⁻¹) compared with $Zn^{2+}$–$H_2O$ complex (−104.54 kJ mol⁻¹), as shown in Supplementary Fig. S12.

To ascertain the coordination number of anions and solvents in the solvation sheath of TMP–40, molecular dynamic (MD) simulations were carried out. The numbers of $Zn^{2+}$, OTf⁻, TMP and $H_2O$ in the hybrid electrolytes are summarized in Supplementary Table S2. As shown by the snapshots from the simulated solvation structure (Supplementary Figs. S13 and S14), some water molecules are squeezed out of the $Zn^{2+}$ solvation shell in the TMP–40 electrolyte and partially replaced with TMP solvent and OTf⁻ anions. According to the radial distribution functions (RDF) in Fig. 2c, the Zn–O peak in OTf⁻, TMP and $H_2O$ correspond to the distance of 0.19, 0.25 and 0.23 nm, respectively, further validating that the OTf⁻, TMP and $H_2O$ molecules incorporate into the first solvation shell of $Zn^{2+}$. Accordingly, the respective coordination number was calculated to be 0.84, 0.14, and 5.02, constituting a CIP-type solvation shell of $Zn^{2+}[H_2O]_{5.02}[TMP]_{0.14}[OTf^-]_{0.84}$, which favors the in situ formation of SEI on Zn surface through reductive decomposition. Moreover, the solvation energy of $Zn^{2+}$ in $Zn(OTf)(TMP)(H_2O)_4$ cluster and solvation sheath geometries of $Zn(OTf)(TMP)(H_2O)_4$ cluster were simulated using high-level density functional theory (DFT) and PCFF-INTERFACE force fields, respectively, to verify the accuracy of MD simulations (Supplementary Fig 15). The $Zn^{2+}$ desolvation energy of TMP–0 and TMP–40 can be obtained by extracting the respective $R_{ct}$ at different temperatures before cycling[36,37], where no SEI or passivation film is formed on the Zn surface. As shown in

Supplementary Fig. S16, the addition of TMP causes a slight increase in the energy barrier for $Zn^{2+}$ dissociation, largely ascribed to the strong interaction of $Zn^{2+}$–TMP and $Zn^{2+}$–OTf⁻, which in turn allows for the stepwise formation of $ZnF_2$–$Zn_3(PO_4)_2$.

With the TMP–40 as the optimal electrolyte formulation, the linear sweep voltammetry (LSV) measurements were first performed to examine the electrochemical stability. It was found that the TMP–40 electrolyte can effectively prevent Zn surface corrosion and suppress water decomposition over a wide electrochemical window (Supplementary Figs. S17 and S18). Moreover, the cathodic peak at -0.05 to 0.20 V gradually decreases as the cycling proceeds and almost disappears after 5 cycles, which remains almost unchanged even after 20 cycles (Supplementary Fig. S19), indicating the as-formed SEI remains stable after 5 cycles. The characteristic peaks corresponding to the zinc triflate hydroxide hydrate ($Zn_xOTf_y(OH)_{2x−y}·nH_2O$, ZOTH) were detected in the X-ray diffraction (XRD) pattern of the Zn surface in TMP–0 after 40 cycles of stripping/plating (Supplementary Fig. S20), which can seriously restrict the transport of $Zn^{2+}$ and lead to dendrite growth[32,38]. In contrast, no by-product was observed in TMP–40.

X-ray photoelectron spectroscopy (XPS) with Ar ion sputtering was further employed to determine the depth distribution of compositions in SEI formed on the Zn surface. As shown in Fig. 2d, the top SEI layer (before sputtering) is rich in −$CF_3$ species (-688.8 eV) and inorganic $ZnF_2$ (-684.1 eV) with a tiny amount of $Zn_3(PO_4)_2$ (-134.3 eV). Accordingly, the lattice fringes in the high-resolution transmission electron microscopy (HRTEM) corresponding to the planes of $ZnF_2$ and $Zn_3(PO_4)_2$ were clearly observed with uniform distribution (Supplementary Fig. S21). Based on the previous reports[29,32], the −$CF_3$ species arises from either the incomplete reduction of OTf⁻ or the residual salt on Zn surface, while $ZnF_2$ and $Zn_3(PO_4)_2$ are attributed to the decomposition product of $Zn^{2+}$–OTf⁻ and $Zn^{2+}$–TMP complexes. As the sputtering continues, the peak intensity of $Zn_3(PO_4)_2$ distinctly

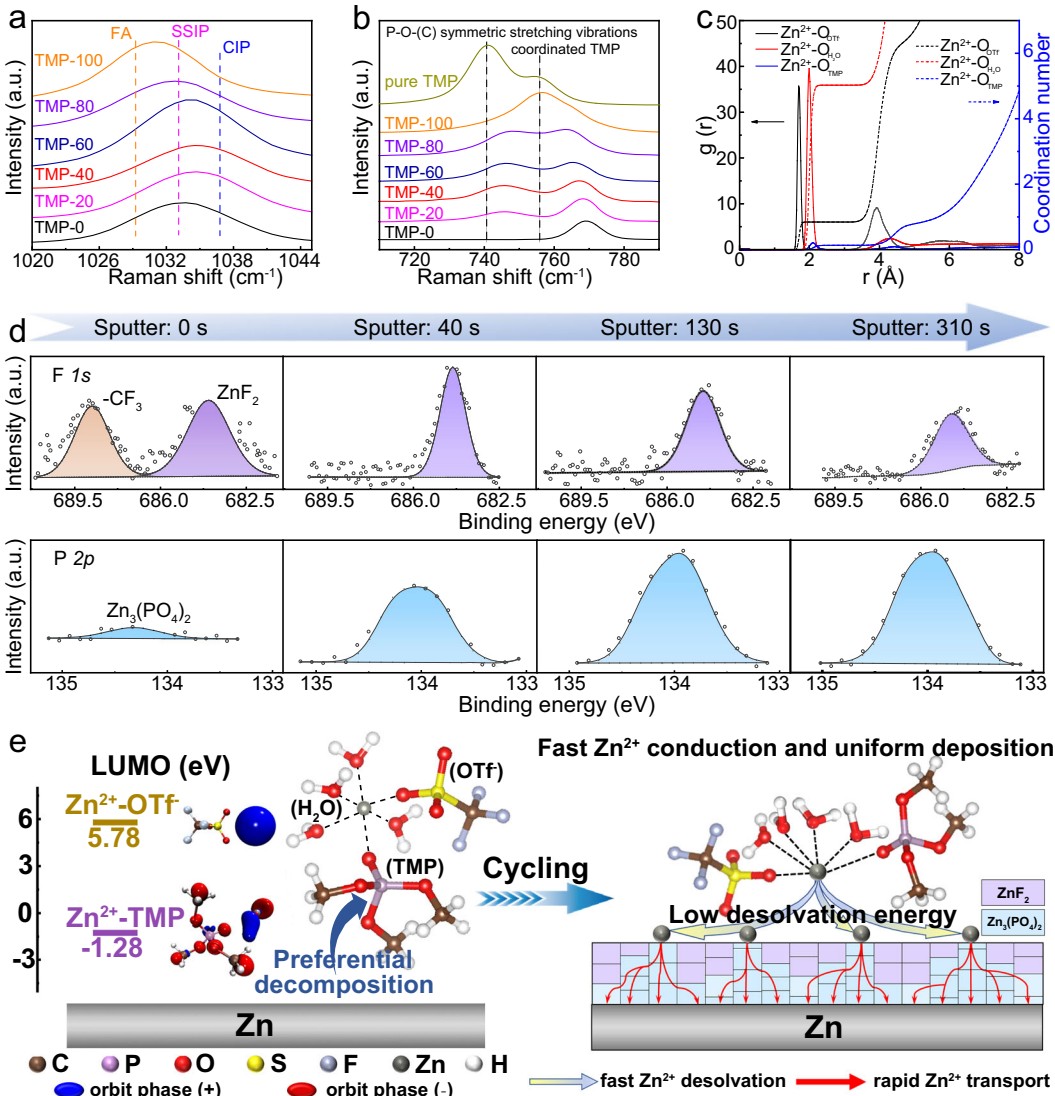

**Fig. 2 | Characterizations of Zn²⁺ solvation structure and the in situ SEI components. a** $SO_3$ stretching mode in Raman spectra for different hybrid electrolytes. FA, SSIP and CIP stand for free anion (OTf⁻), solvent-separated ion pairs (Zn²⁺–(H₂O)ₓ(TMP)ᵧ–OTf⁻) and contact ion pairs (Zn²⁺–OTf⁻), respectively. **b** The P–O–(C) symmetric stretching vibration of different hybrid electrolytes. **c** Profiles of radial distribution function g(r) (solid line) and its integral that represents the coordination number (dash-dotted line) of Zn²⁺ with O in different components of TMP-40 electrolyte. **d** XPS F *1s* and P *2p* spectra with increasing sputtering time. **e** Schematic illustration of formation process (left) and action mechanism (right) about ZnF₂-Zn₃(PO₄)₂ interlayer.

becomes stronger along with the decrease in ZnF₂ peak (Fig. 2d and Supplementary Fig. S22). After 310 s of sputtering, the Zn₃(PO₄)₂ gradually becomes the major composition in the SEI. In sharp contrast, no F or P signals related to ZnF₂ or Zn₃(PO₄)₂ was observed in TMP-0 (Supplementary Fig. S23). The XPS analyses provide strong evidence that the TMP-40 electrolyte favors the in situ formation of gradient interlayer on the Zn surface, where ZnF₂ and Zn₃(PO₄)₂ dominate the surface layer and inner layer, respectively. Furthermore, the cross-sectional information of the as-formed SEI in the TMP-40 electrolyte was collected through a focused ion beam (FIB) cutting technique. As shown in Supplementary Fig. S24, the gradient distribution of ZnF₂ and Zn₃(PO₄)₂ can be clearly revealed by both EDS mappings and line scans of F and P elements, further evidencing the formation of gradient SEI. By contrast, no obvious P or F signal was detected in the case with TMP-0 electrolyte (Supplementary Fig. S25).

To gain an insight into the formation process of gradient ZnF₂–Zn₃(PO₄)₂ SEI, we employed DFT calculations to compare the LUMO energy level of Zn²⁺–TMP and Zn²⁺–OTf⁻ in the solvation shell. Obviously, the LUMO energy level of Zn²⁺–TMP is much lower than that

of Zn²⁺–OTf⁻ (Fig. 2e left and Supplementary Fig. S26). Meanwhile, the coordinated TMP in the Zn²⁺-solvation sheath is farther from the Zn²⁺ than OTf⁻ (Fig. 2c). Collectively, the coordinated TMP can preferentially accept electrons from the Zn metal to be reduced into Zn₃(PO₄)₂, followed by the decomposition of OTf⁻ into ZnF₂, forming gradient SEI with Zn₃(PO₄)₂ at the bottom and ZnF₂ on the top, as schematically shown in Fig. 2e Right. The outer ZnF₂ promotes the Zn²⁺ desolvation, and inner Zn₃(PO₄)₂ facilitates rapid Zn²⁺ transport across the SEI, respectively, for ZMBs to stably work at low temperatures, as discussed later.

### The kinetics behavior of bivalent Zn²⁺ on the electrode/electrolyte interface

The temperature-dependent electrochemical impedance spectroscopy (EIS) measurements of Zn||Zn cells were performed at temperatures ranging from 20 °C to −30 °C in TMP-0 and TMP-40 after 40 cycles (Supplementary Fig. S27), where a dense SEI should be formed in TMP-40. The charge transfer resistance ($R_{ct}$) and the resistance associated with Zn²⁺ crossing SEI ($R_{SEI}$) can be extracted from the

semicircles in the mid-frequency region and the high-frequency region, respectively[26–28]. By Arrhenius-fitting $R_{ct}$ and $R_{SEI}$ over 1000/ T, the activation energy of each interface process was obtained, as shown in Fig. 3a, b. Compared with TMP-0, the desolvation energy of $Zn^{2+}$ in the TMP-40 was greatly reduced (70.2 vs 54.8 kJ $mol^{-1}$), indicating the outer $ZnF_2$ facilitate $Zn^{2+}$ desolvation, agreeing with the reported results[39,40]. Moreover, the activation energy for $Zn^{2+}$ transport through the SEI in TMP-40 ($E_{a,SEI}$ = 52.7 kJ $mol^{-1}$) is significantly lower than in TMP-0 ($E_{a,SEI}$ = 64.3 kJ $mol^{-1}$). This can be well explained by the DFT calculation results (Supplementary Fig. S28) that the $Zn_3(PO_4)_2$ delivers a much smaller migration energy barrier for $Zn^{2+}$ (0.38 eV) (Fig. 3d) and higher affinity with $Zn^{2+}$ (−1.15 eV) (Fig. 3c) compared to those of $ZnF_2$ (1.12 eV for $Zn^{2+}$ transport and weak binding energy of −0.85 eV with $Zn^{2+}$). That is, the rich $Zn_3(PO_4)_2$ in the inner SEI serves as the dominant channel for the desolvated $Zn^{2+}$ across the SEI to deposit on the Zn surface, which can facilitate fast $Zn^{2+}$ conduction to minimize voltage polarization under cold environments[41,42]. Moreover, we prepared SEI containing single $ZnF_2$ or $Zn_3(PO_4)_2$ to confirm their respective roles (Supplementary Fig. S29). Besides, the interface impedance of Zn||Zn cells in the TMP-40 electrolyte can remain stable after 500 cycles, indicating the stable interface due to the formation of $ZnF_2$–$Zn_3(PO_4)_2$ interlayer (Fig. 3e). In sharp contrast, the TMP-0 electrolyte shows a sharp decrease in the interfacial impedance after 100 cycles, possibly due to the dendrite growth that causes the cell short circuit (Fig. 3f)[43].

We further studied the kinetics of $ZnF_2$–$Zn_3(PO_4)_2$ gradient SEI at −30 °C. SEI or passivation film can be grown on Zn metal surfaces using TMP-0 and TMP-40 electrolytes at room temperature, which are denoted as ZOTH@Zn and $ZnF_2$–$Zn_3(PO_4)_2$@Zn, respectively. As shown in Supplementary Fig. S30, the $ZnF_2$–$Zn_3(PO_4)_2$@Zn symmetric cells exhibit lower voltage polarization at different current densities, implying that the gradient SEI reduces the nucleation barrier and improves Zn affinity. Meanwhile, $ZnF_2$–$Zn_3(PO_4)_2$@Zn manifests a lower charge transfer resistance (Fig. 3g), which is expected to facilitate Zn nucleation. Moreover, $ZnF_2$–$Zn_3(PO_4)_2$@Zn exhibits a higher $Zn^{2+}$ migration number (0.87 vs. 0.46) and larger peak current density (-0.5 vs. 0.15 mA $cm^{-2}$) as compared to those of ZOTH@Zn (Fig. 3h, i and Supplementary Fig. S31). These results fully demonstrate that the Zn electrode modified with $ZnF_2$–$Zn_3(PO_4)_2$ gradient SEI exhibits superior kinetics at low temperatures.

In addition, high mechanical integrity is indispensable for SEI to ensure long-term cycling and high-capacity plating. To this end, in situ optical microscopy was performed to observe the morphology evolution at different plating stages. As shown in Supplementary Fig. S32, the TMP-40 electrolyte enables dense deposition without dendrite formation during the whole deposition process and can maintain a smooth surface even at a colossal loading of 50 mAh $cm^{-2}$, while for the case of TMP-0, uneven spots appear at 10 mAh $cm^{-2}$ and gradually evolve into discontinuous islands as the plating capacity increases, which is consistent with the scanning electron microscopy (SEM)

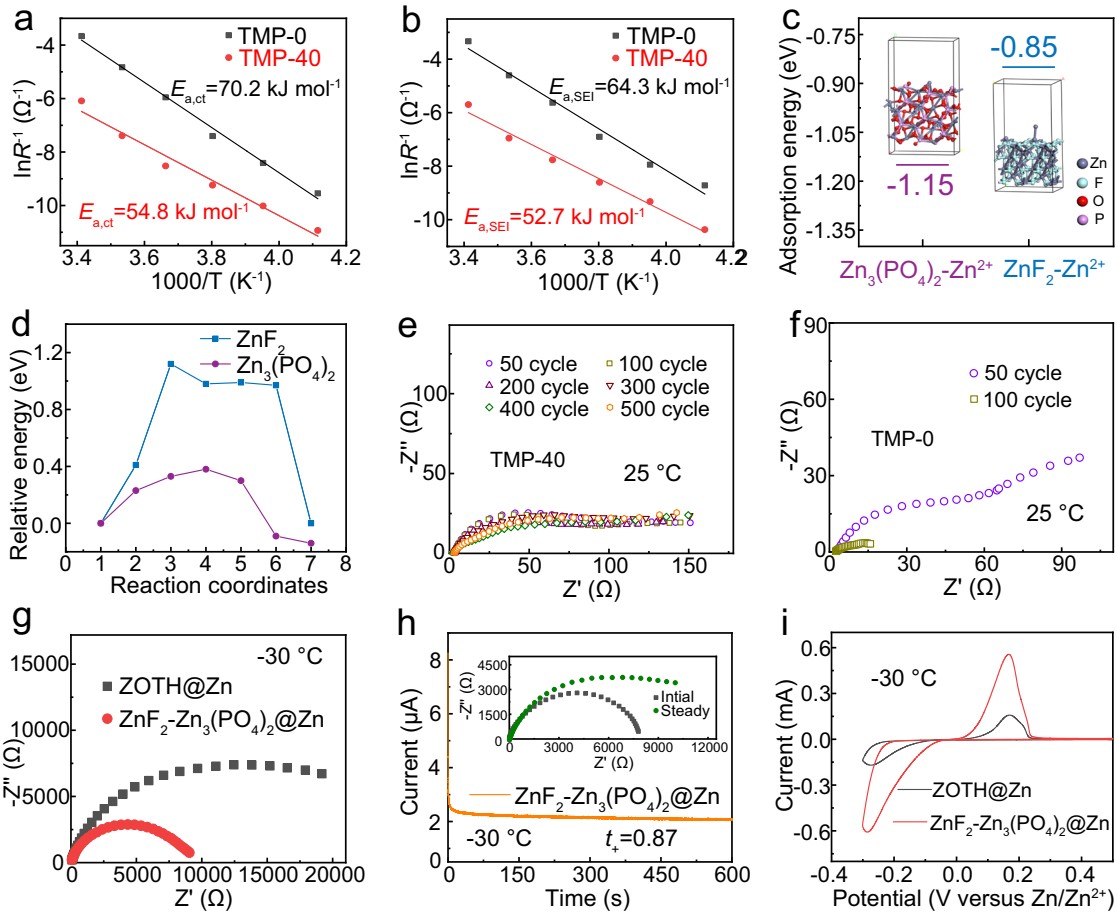

**Fig. 3 | Properties and functions study of SEI in situ formed on Zn surface in the electrolyte of TMP-40.** Arrhenius curves and the activation energies of **a** $R_{ct}$ and **b** $R_{SEI}$ derived from the Nyquist plots of Zn||Zn cells with TMP-0 or TMP-40 electrolyte after 40 cycles. **c** Adsorption energy of $Zn^{2+}$ with $ZnF_2$ and $Zn_3(PO_4)_2$. **d** Migration energy barrier of $Zn^{2+}$ across $ZnF_2$ and $Zn_3(PO_4)_2$ interlayer. Electrochemical impedance spectra of Zn||Zn symmetric cells in **e** TMP-40 electrolyte and **f** TMP-0 electrolyte after different cycles at 25 °C. **g** Electrochemical impedance spectra comparison between ZOTH@Zn and $ZnF_2$–$Zn_3(PO_4)_2$@Zn in symmetric cells with TMP-40 electrolyte at −30 °C. **h** $Zn^{2+}$ transference number test for $ZnF_2$–$Zn_3(PO_4)_2$@Zn in symmetric cells at −30 °C. **i** CV comparison between ZOTH@Zn||Ti and $ZnF_2$–$Zn_3(PO_4)_2$@Zn||Ti at −30 °C.

results (Supplementary Fig. S33). The striking contrast in morphology evolution demonstrates that the gradient SEI can effectively suppress the dendrite growth and is highly stable to accommodate the high-loading Zn plating, largely ascribed to the strong bulk modulus of $Zn_3(PO_4)_2$[42]. Taken together, we reasonably conclude that the rich $ZnF_2$ on the top layer of SEI favors the desolvation of $Zn^{2+}$ and the robust $Zn_3(PO_4)_2$ predominating the inner SEI layer facilitates rapid $Zn^{2+}$ transport. With these admirable characteristics, it is anticipated that the as-formed gradient $ZnF_2–Zn_3(PO_4)_2$ SEI can guarantee stable and long-term cycling of Zn metal at low temperatures.

### Electrochemical performance of Zn metal anodes under harsh conditions

It is worth mentioning that these features of the gradient SEI can allow the symmetric Zn cells to stably cycle in the TMP−40 electrolyte at a high current density of 5 mA cm⁻² at 25 °C and 45 °C (Fig. 4a and Supplementary Fig. S34). Impressively, the cell can stably cycle even if the current density reaches 50 mA cm⁻² at 25 °C (Supplementary Fig. S35). In sharp contrast, the cells in TMP−0 quickly failed due to severe Zn dendrite formation and aggravated side reactions. This also manifests that the gradient $ZnF_2–Zn_3(PO_4)_2$ SEI is stable against the high-temperature and high-rate cycling, underscoring the significance of SEI formation on the interface. Then the galvanostatic cycling stability of Zn metal in the TMP−40 electrolyte was studied at low temperatures with different rates. When the operation temperature was fixed at −30 °C, the cell in TMP−40 exhibited a stable voltage profile at 2 mA cm⁻² with an ultralong cycling life of up to 3600 h, which is around 40-fold improvement (Fig. 4b). With the superior kinetics in

the electrolyte/electrode interface, the TMP−40 electrolyte enables the symmetric cells to operate over long-term cycles at high rates ranging from 5 to 15 mA cm⁻² through a transient activation (Supplementary Fig. S36).

Then the cycling temperature was decreased to −50 °C. Not surprisingly, the cell in TMP−0 cannot work due to the electrolyte solidification (Fig. 4c). While with a high content of TMP (e.g., TMP−60 or TMP−80), the cells have a large charge transfer impedance, which results in severe polarization and huge overpotential (Supplementary Fig. S37). Comparatively, the TMP−40 electrolyte achieves an ultralong lifespan at 0.4 mA cm⁻² and 0.4 mAh cm⁻² without obvious fluctuation in overpotential over 7000 h (-10 months). As the discharge depth was increased to 1 mAh cm⁻², it was amazing to find that the cell can still maintain impressive stability over 6000 h (Supplementary Fig. S38). Meanwhile, the cell can operate normally at a current density of up to 2 mA cm⁻² (Supplementary Fig. S39). These observations convectively demonstrate that the as-formed gradient $ZnF_2–Zn_3(PO_4)_2$ SEI with the unique configurations can accelerate the rapid $Zn^{2+}$ desolvation and conduction at low temperatures, which guarantees stable cycling with a negligible polarization under rather extreme conditions (low temperatures with high rates). Notably, the Zn‖Zn symmetric cells in TMP−40 achieve a rather competitive cumulative capacity over a wide temperature range (Fig. 4d and Supplementary Table S3), far outperforming those of reported low-temperature aqueous Zn metal anodes[8,13,17,19,44,45]. Both top and side views reveal that the surface of the Zn electrode after 100 plating/stripping cycles in TMP−40 is highly homogeneous and tightly packed (Fig. 4e), while the one with TMP−0 exhibits severe cracks (Fig. 4f), indicating the gradient SEI layer can

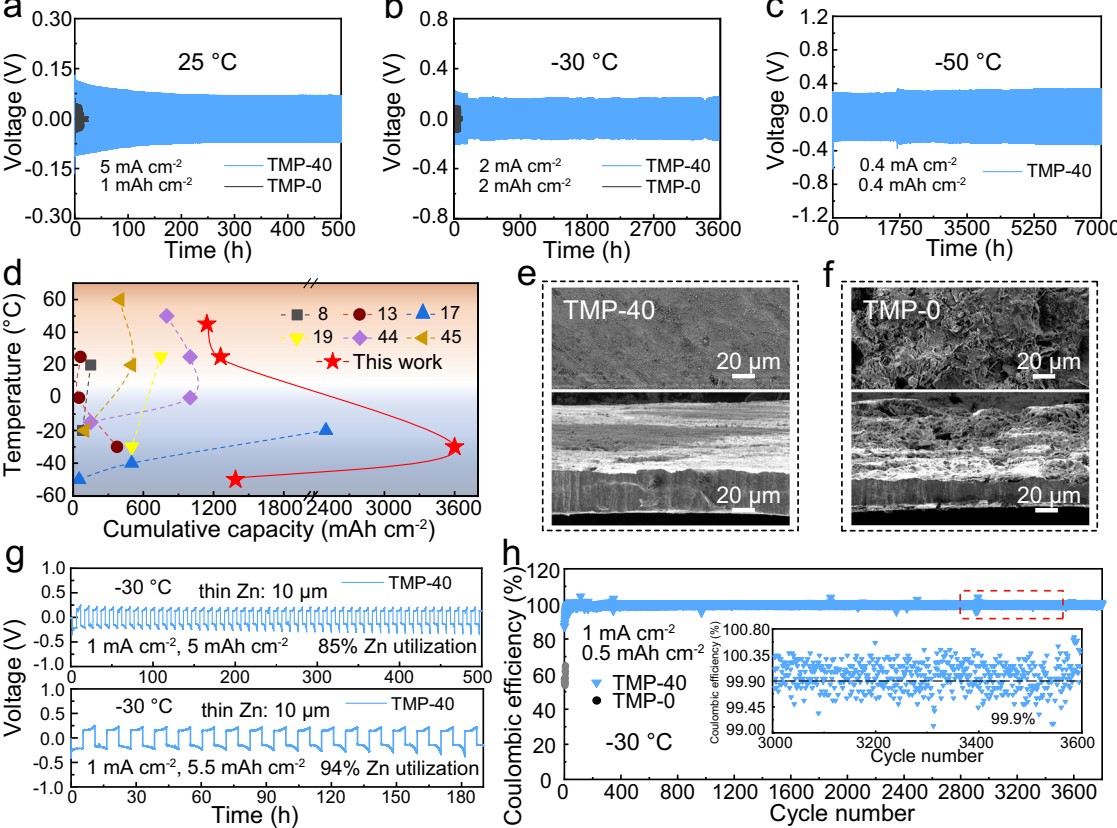

**Fig. 4 | Electrochemical test of symmetric Zn cells under harsh conditions.** Galvanostatic cycling stability of symmetric Zn cells with TMP−40 and TMP−0 electrolytes, respectively, at a temperature of **a** 25 °C, **b** −30 °C, and **c** −50 °C. **d** Comparison of the cumulative capacity at wide temperatures in our work with other reported results. Top-view (top) and cross-section (bottom) SEM images of

symmetric Zn cells after 100 cycles with electrolytes of **e** TMP−40 and **f** TMP−0. **g** Galvanostatic cycling stability of symmetric Zn cells with TMP−40 electrolyte under 85% and 94% of Zn utilization at −30 °C. **h** Long-term Zn plating/stripping Coulombic efficiency in TMP−40 and TMP−0 electrolytes at −30 °C. Insets: the magnified views of selected cycles.

effectively suppress the Zn dendrite growth. Moreover, the cycling stability of Zn plating and stripping under actual conditions was evaluated using 10 μm thickness Zn foil (5.85 mAh cm$^{-2}$) at −30 °C. With a capacity of 4 mAh cm$^{-2}$ corresponding to the Zn utilization rate of 68%, the symmetric cell with TMP−40 electrolyte exhibits a highly stable voltage profile over 1800 h (Supplementary Fig. S40). As the Zn utilization rate was increased to 85% and 94%, it is amazing to find that the cells can still maintain high stability over 500 h with 5 mAh cm$^{-2}$ and over 180 h with 5.5 mAh cm$^{-2}$ (Fig. 4g). These results demonstrate the viability of the gradient SEI in stabilizing ZMBs under practical and harsh conditions.

The reversibility of Zn plating/stripping was further studied by calculating the Coulombic efficiency of Zn metal onto titanium (Ti) substrate. At −30 °C with a current density of 1 mA cm$^{-2}$ and capacity of 0.5 mAh cm$^{-2}$, the Coulombic efficiency in TMP−40 electrolyte quickly increases to 99% within 20 cycles and stabilizes at 99.9% over 3800 cycles along with flat voltage profiles (Fig. 4h and Supplementary Fig. S41). Conversely, the cell with TMP−0 electrolyte exhibits a low Coulombic efficiency of around 60% and quickly short-circuited at the ninth cycle due to dendrite formation. These contrasts can be maximized at different temperatures ranging from 45 °C to −50 °C (Supplementary Fig. S42), well illustrating that the gradient SEI layer is stable against the severe side reaction and maintains favorable kinetics at low temperatures.

The superior stability of ZMBs at subzero temperature drops strong hint that the temperature may influence the interphase evolution even with the same electrolyte components. To this end, we compared the character of SEI formed at room temperature (25 °C) and −30 °C, denoted as ZnF$_2$−Zn$_3$(PO$_4$)$_2$@Zn (25 °C) and

ZnF$_2$−Zn$_3$(PO$_4$)$_2$@Zn (−30 °C). Firstly, we evaluated the electrochemical windows of TMP−40 electrolyte at different temperatures using LSV (Fig. 5a−c). The low-temperature environment can simultaneously widen the anodic and cathodic limits of the electrolyte due to the inhibition of water activity. Further study shows that a low-temperature environment decreases the onset potential of HER from 271 mV down to −317 mV (versus Zn/Zn$^{2+}$). The hydrogen evolution current of 30.89 μA cm$^{-2}$ at 0.5 V was eliminated (red arrows, Fig. 5b). Meanwhile, the oxygen evolution reaction (OER) was suppressed with an onset potential increasing to 2.7 V from 2.0 V (Fig. 5c). Overall, an electrochemical window of ~3.0 V was achieved with the TMP−40 electrolyte at −30 °C, much wider than its at room temperature (~1.7 V). Benefiting from the suppressed side effects, the SEI generated at −30 °C is much denser and more uniform (Fig. 5d) as compared to that formed at room temperature (Fig. 5e). Moreover, the EIS spectrum of ZnF$_2$−Zn$_3$(PO$_4$)$_2$@Zn (−30 °C) shows a much lower charge transfer resistance at −30 °C (Fig. 5f), indicating that the gradient SEI formed at subzero temperature has rapid diffusion kinetics, which is beneficial for stable cycling under cold conditions. Accordingly, the symmetric cell with ZnF$_2$−Zn$_3$(PO$_4$)$_2$@Zn (−30 °C) exhibits a much better rate performance, in which the voltage hysteresis remains stable at current densities ranging from 1 to 30 mA cm$^{-2}$ and discharge capacities ranging from 1 to 30 mAh cm$^{-2}$ under 25 °C (Fig. 5g). Moreover, it can be stably cycled over 140 h at a high current density of 20 mA cm$^{-2}$ and high discharge depth of 20 mAh cm$^{-2}$ (Supplementary Fig. S43). On the contrary, the cell with ZnF$_2$−Zn$_3$(PO$_4$)$_2$ (25 °C) shows larger voltage polarization and short circuits at a discharging depth of 20 mAh cm$^{-2}$. These results indicate that the operation temperature plays an important role in determining the final morphology of SEI and relevant

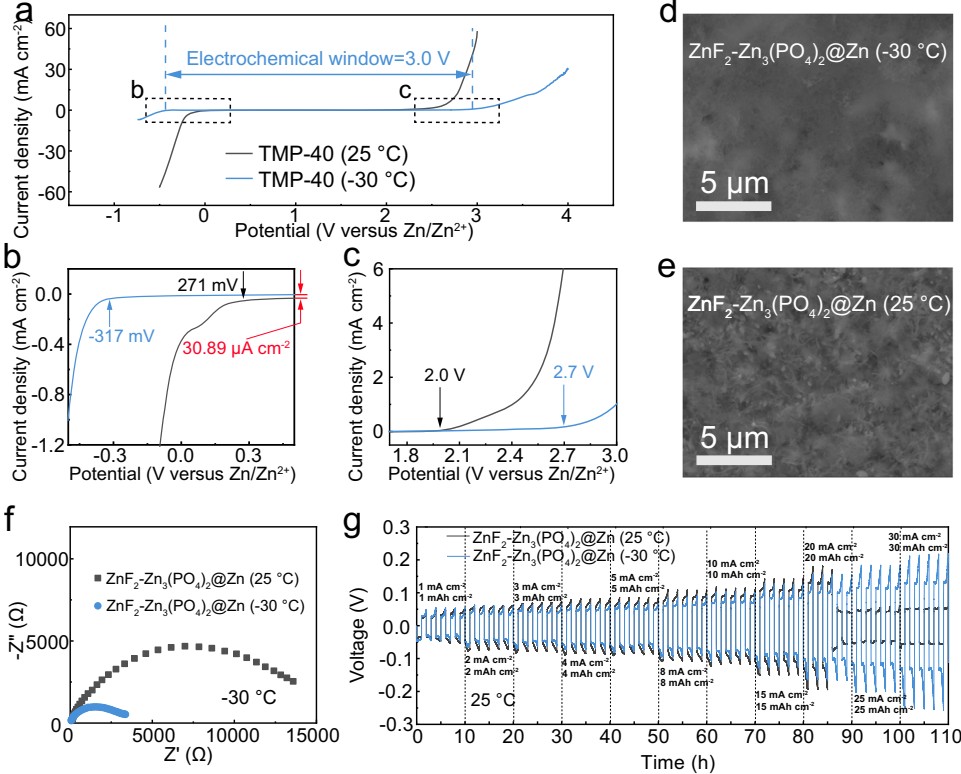

**Fig. 5 | Study of temperature effect on SEI formation. a** The electrochemical window of TMP−40 electrolyte measured using polarization scanning at 0.5 mV s$^{-1}$ on non-active Ti electrodes under different temperatures. The corresponding amplification curves at **b** −0.5 to 0.5 V and **c** 1.7 to 3.0 V. SEM characterizations of **d** ZnF$_2$−Zn$_3$(PO$_4$)$_2$@Zn (−30 °C) and **e** ZnF$_2$−Zn$_3$(PO$_4$)$_2$@Zn (25 °C). **f** Electrochemical impedance spectra comparison between ZnF$_2$−Zn$_3$(PO$_4$)$_2$@Zn

(25 °C) and ZnF$_2$−Zn$_3$(PO$_4$)$_2$@Zn (−30 °C) in symmetric cells at −30 °C. **g** Comparison of stability of Zn$^{2+}$ deposition/stripping between ZnF$_2$−Zn$_3$(PO$_4$)$_2$@Zn||ZnF$_2$−Zn$_3$(PO$_4$)$_2$@Zn (25 °C) and ZnF$_2$−Zn$_3$(PO$_4$)$_2$@Zn|| ZnF$_2$−Zn$_3$(PO$_4$)$_2$@Zn (−30 °C) at 25 °C under different current densities and discharge capacities.

$Zn^{2+}$ transport behavior, further affecting the low-temperature performance.

## Electrochemical performance of full cells under practical conditions

To evaluate the practical applications of TMP−40 electrolyte, KVOH (Supplementary Fig. S44) was employed as a cathode to pair with Zn metal for full cells due to its superior kinetics[46]. The Zn−KVOH full cell with TMP−40 electrolyte delivers an initial capacity of 329.1 mAh g⁻¹ at 5 A g⁻¹ and retains a capacity of 323.6 mAh g⁻¹ after 2300 cycles at room temperature (Fig. 6a), corresponding to a capacity retention of 98.3%. In contrast, the cell with TMP−0 electrolyte exhibits a slightly higher initial capacity of 344.4 mAh g⁻¹ at 5 A g⁻¹, possibly due to higher ionic conductivity, but quickly dropped to 211 mAh g⁻¹ after 500 cycles, along with a larger polarization in the charge−discharge voltage profiles (Supplementary Fig. S45). The significant performance improvement is also revealed with high loadings of KVOH (6.37 and 17.6 mg cm⁻²), at various rates (1, 2, and 10 A g⁻¹) or even at a higher temperature (45 °C, 5 A g⁻¹), as shown in Supplementary Figs S46–48. Impressively, with a high areal loading of KVOH up to 17.6 mg cm⁻², the full cell still maintains an areal capacity of 4.37 mAh cm⁻² after 100 cycles, which meets the requirements of a typical commercial Li-ion battery (4.0 mAh cm⁻²)[47]. In view of the inspiring performance, we further evaluated the application of TMP−40 electrolyte in practical situations with lean electrolyte and low Zn excess. As shown in Fig. 6b, when the KVOH loading increases to 33.75 mg cm⁻², the cell still delivers an initial areal capacity of 9.42 mAh cm⁻² with lean E/C (6.76 μL mAh⁻¹, the ratio of electrolyte volume to capacity) ratio and low N/P (3.1, the ratio of negative to positive). The corresponding energy density is calculated to be 251.2 Wh kg⁻¹ (based on the KVOH mass) with a high capacity retention of 93.3% after 50 cycles. Moreover, we validated the universality of TMP−40 electrolyte using high-voltage cathode $MnO_2$. As displayed in Supplementary Fig. S49, the Zn−$MnO_2$ full cell delivers a high areal capacity of 3.68 mAh cm⁻² at a low N/P (~3.2) and stably cycles over 100 cycles. The outstanding performance can be ascribed to the superior kinetics and great robustness of the $ZnF_2$−$Zn_3(PO_4)_2$ SEI that allows large amounts of $Zn^{2+}$ to repeatedly strip and plate.

## Electrochemical performance of full cells at low temperature

By virtue of the favorable SEI formation, the TMP−40 electrolyte can enable the Zn−KVOH full cells to sustain remarkably long lifespan and prominent stability at subzero temperatures. Specifically, when the temperature was decreased to −30 °C (Supplementary Fig. S50), the discharge capacity of the cell in TMP−40 remains 120.6 mAh g⁻¹ after 4000 cycles at 1 A g⁻¹, far exceeding that without TMP. Even at a higher rate of 2 and 5 A g⁻¹, the 40% of TMP addition can prompt the cell to charge/discharge reversibly over 10,000 cycles and 1500 cycles, respectively. In stark contrast, the cell with TMP−0 electrolyte failed to work, underscoring the critical role of the gradient SEI formed in the TMP−40 electrolyte. These contrasts are more evident at −50 °C, where the cell with TMP−0 cannot provide any capacities

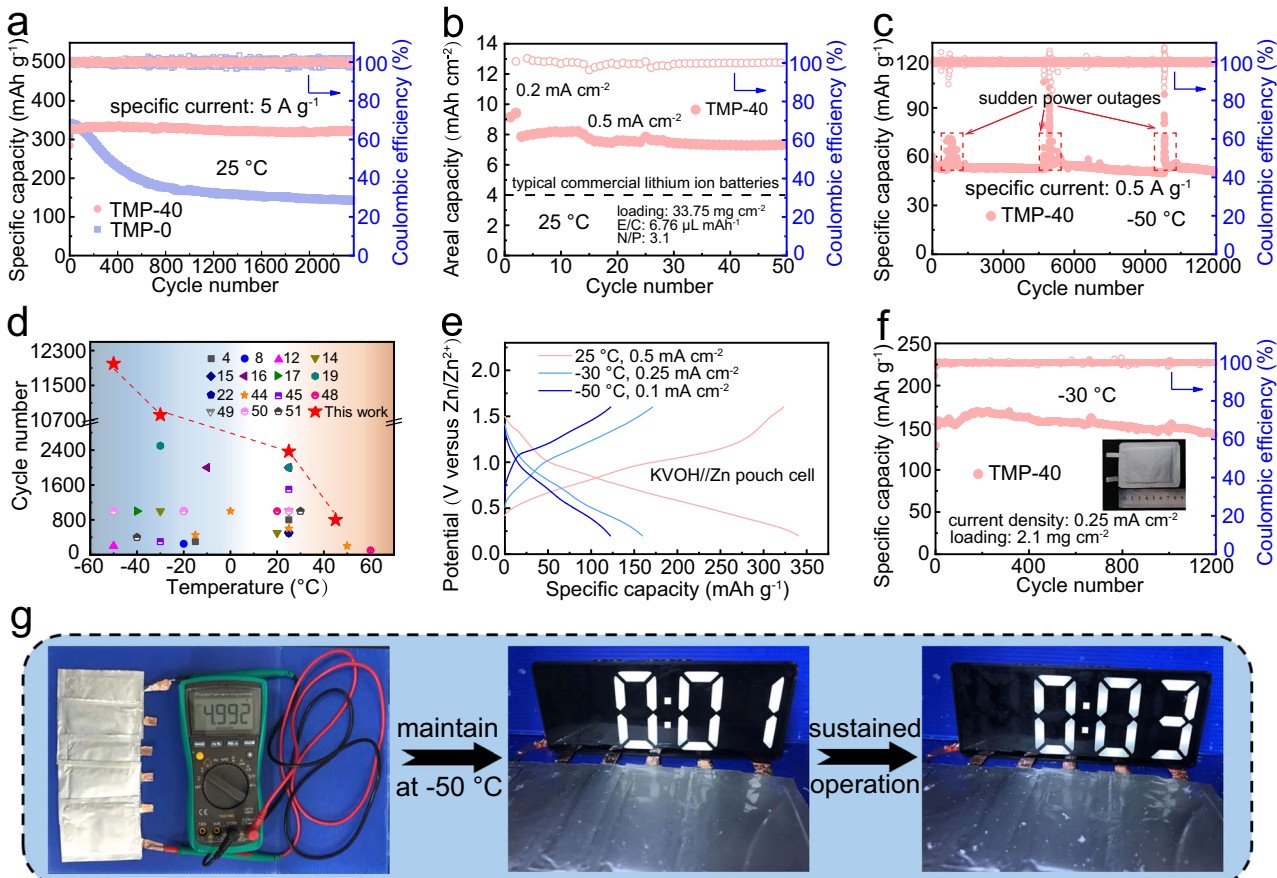

**Fig. 6 | Electrochemical performance of Zn−KVOH full cells. a** Cycling performance of Zn−KVOH cells in the electrolyte of TMP−0 and TMP−40 at 25 °C. **b** Zn−KVOH full cell test under practical conditions with TMP−40 electrolyte. **c** Cycling performance of Zn−KVOH cells with TMP−40 electrolyte at −50 °C. **d** Comparison of the cycling performance of full cells at wide temperatures achieved in this work with those reported ones in the literature. **e** Discharge−charge curves of Zn−KVOH pouch cell with TMP−40 electrolyte at 25 °C, −30 °C, and −50 °C. **f** Cycling performance of pouch cell with TMP−40 electrolyte at −30 °C. Inset: Photograph of the pouch cell. **g** Pictures of the five series connected pouch cells to drive the calculagraph working for more than 3 min at −50 °C.

due to the electrolyte solidification. Comparatively, the TMP−40 electrolyte can render the full cell to deliver a stable capacity of 50.8 mAh g$^{-1}$ at 0.5 A g$^{-1}$ over 12,000 cycles (Fig. 6c). The fluctuations at the 800th, 5000th, and 9800th cycle are ascribed to the sudden power outage during the long-term test, which also confirms the good temperature adaptivity of the gradient SEI formed in TMP−40. Compared with the published works on electrolyte modification for aqueous Zn batteries, this work is undoubtedly prominent in achieving long-cycle stability over a wide temperature range (Fig. 6d and Supplementary Tables S4 and S5)[4,8,12,14–17,19,22,44,45,48–51]. Post-mortem analyses show that the Zn metal anode after 1000 cycles in the TMP−40 electrolyte maintains a much flatter and denser surface (Supplementary Fig. S51). The significant performance improvement achieved in the TMP−40 electrolyte at low temperatures strongly validates the gradient SEI with favorable kinetics can effectively suppress Zn dendrite growth and ensure superb cycling behaviors in full cells under extreme conditions.

The exceptional performance of the ZMBs inspired us to further evaluate the low-temperature performance of pouch cell with a size of 4.3 × 5.6 cm in the TMP−40 electrolyte. As shown in Fig. 6e, a reversible capacity of 340.5, 159.4, and 122.2 mAh g$^{-1}$ is achieved at 25 °C (room temperature), −30 °C and −50 °C, respectively. Notably, the pouch cell achieves superior cycling stability at a low temperature of −30 °C with a high capacity retention of 88.6% after 1200 cycles at 0.25 mA cm$^{-2}$ (Fig. 6f). Even at −50 °C, it can stably cycle for 180 cycles with nearly 100% of capacity retention (Supplementary Fig. S52). Moreover, five series connected pouch cells with a voltage of ~5 V can drive an electro-calculagraph normally working for more than 3 min at an extreme temperature of −50 °C (Fig. 6g and Supplementary Fig. S53), further demonstrating its promising potential in the practical applications under harsh conditions.

## Discussion

In summary, we have successfully in situ constructed gradient SEI layer on the Zn metal surface to enable long-cycling and dendrite-free ZMBs at low temperatures by introducing TMP into the aqueous electrolyte. It is found that 40% of TMP addition can regulate the Zn$^{2+}$ solvation sheath and promote the formation of ZnF$_2$−Zn$_3$(PO$_4$)$_2$ gradient SEI due to the difference in chemical reactivity of TMP cosolvents and OTf$^-$ anions, which effectively suppress the parasitic reaction and Zn dendrite growth. Moreover, a combination study of computational and experimental characterizations reveals the outer ZnF$_2$ can facilitate Zn$^{2+}$ desolvation and the inner Zn$_3$(PO$_4$)$_2$ can serve as dominant channels for fast Zn$^{2+}$ conduction, synergistically leading to continuous cycling under cold environments. Remarkably, the Zn symmetric cells reach superior stability at −50 °C with a lifespan of 7000 h and achieve a high Zn utilization rate of 94% at −30 °C. Furthermore, the Zn−KVOH full cell delivers a stable capacity of 50.8 mAh g$^{-1}$ at 0.5 A g$^{-1}$ over 12,000 cycles at −50 °C. Full cells with lean electrolyte and low Zn excess also demonstrate the practical feasibility. This work provides a feasible route to achieve highly stable ZMBs under extreme conditions from the perspective of interlayer regulation.

## Methods

### Electrolyte preparation

Zinc trifluoromethanesulfonate (Zn(OTf)$_2$, >99%) and trimethyl phosphate (TMP, 98%) were purchased from Adamas and Sigma-Aldrich, respectively. A series of electrolytes was formulated by dissolving 2 M Zn(OTf)$_2$ in deionized (DI) water with different volume ratios of TMP from 0%, 5%, 10%, 20%, 40%, 60%, 80% to 100%. The corresponding electrolyte is marked as TMP−0, TMP−5, TMP−10, TMP−20, TMP−40, TMP−60, TMP−80 and TMP−100, respectively.

### Synthesis of KVOH cathode

The synthesis of KVOH was based on a previous method, as reported by Gao[46]. Specifically, 0.364 g of vanadium pentoxide (V$_2$O$_5$, 99%, Sigma Aldrich) in 50 mL DI water was mixed with 0.0745 g of potassium chloride (KCl, 99.9%, Sigma Aldrich) in 30 mL DI water, followed by the addition of 1.7 mL hydrogen peroxide (H$_2$O$_2$, 35%, Fisher Scientific) and 0.5 h of stirring. Then the mixture was transferred into a 100 mL Teflon autoclave and heated at 120 °C for 6 h. The green powder was collected by centrifugation, repeated wash with DI water and ethanol, and drying overnight at 60 °C.

### Preparation of ZnF$_2$@Zn and Zn$_3$(PO$_4$)$_2$@Zn

According to the previous report[40], ZnF$_2$@Zn electrode was obtained by chemically treating Zn foil with ammonium fluoride (NH$_4$F) solution. First, 5 mg of NH$_4$F (99.99%, Aladdin) was dispersed in 6 mL of dimethyl sulfoxide (DMSO, 99.9%, Innochem) under vigorous stirring for 3 days and then dropped onto Zn surface, followed by rapidly drying in the glove box at 180 °C for 10 min. The above operation was repeated 10 times to obtain a uniform ZnF$_2$ layer on Zn metal. The Zn$_3$(PO$_4$)$_2$@Zn electrode was obtained through the chemical reaction between metallic Zn and diammonium hydrogen phosphate ((NH$_4$)$_2$HPO$_4$) solution. First, 66 mg of (NH$_4$)$_2$HPO$_4$ (99.9%, Macklin) was dissolved in 10 mL of deionized water to obtain 0.05 M (NH$_4$)$_2$HPO$_4$ aqueous solution. Then the Zn foil was immersed in the as-obtained solution under 5 h of stirring, followed by repeated washing and drying at 80 °C.

### Material characterizations

Differential scanning calorimetry (DSC, METTLER TOLEDO DSC3) was used to measure the freezing point of the hybrid electrolyte at a temperature range from −150 °C to 25 °C, which was carried out in a liquid nitrogen cooling system with a heating rate of 5 °C min$^{-1}$. Fourier Transform Infrared (FT-IR) analysis was conducted on Nicolet iS50. Raman spectroscopy was recorded on LabRAM HR Evolution with an excitation wavelength of 633 nm. The changes in hydrogen bonds were analyzed by nuclear magnetic resonance (NMR, AVANCE III 400 MHz) with deuterated DMSO. Scanning electron microscopy (SEM) images were collected using FEI Microscope (JSM-7900F). The lamella samples were prepared in a scanning electron microscopy equipped with a focused ion beam (FIB-SEM, Crossbeam340) to obtain the cross-sectional information of the as-formed SEI. The current and acceleration voltage of milling and polishing are 0.23 nA, 30 kV and 68 pA, 5 kV, respectively. Meanwhile, the platinum (Pt) was deposited on the electrode surface as a protection layer during sample preparation. Transmission electron microscopy (TEM) and energy dispersive X-ray spectroscopy (EDS) were performed on Talos F200X G2. X-ray diffraction (XRD) measurements were obtained on diffractometer (Smart Lab 9 KW) with a Cu-target X-ray tube ($\lambda$ = 0.154 nm) at 150 mA and 40 kV. The compositions of solid electrolyte interphase (SEI) were determined by X-ray photoelectron spectroscopy (XPS, Axis Ultra DLD) using monochromatic 1486.7 X-ray source. The prominent C $1s$ peak was calibrated to 284.8 eV.

### Electrode preparation

The commercially available Zn foils (>99.99%, Xinyi Metal Materials Co., Ltd) were polished with 8000 mesh sandpaper and used for cell assembly. For the coin cell, the as-prepared KVOH was mixed with super P carbon black (SUPER P, TIMCAL) and polyvinyldifluoride (PVDF, Sinopharm) based on a weight ratio of 7:2:1 (routine test) or 8:1:1 (high loading test), which was dispersed in N-methyl-2-pyrrolidinone (NMP, Sinopharm) to form a slurry. Then the slurry was cast onto carbon paper disks with a diameter of 10 mm and dried overnight at 60 °C in a vacuum oven. The preparation procedures of

the KVOH cathode for punch cells were similar to the above description, except the electrode size is 4.3 × 5.6 cm.

## Electrochemical measurements

2032-type coin cells were assembled to measure the Coulombic efficiency and cycling stability of Zn metal anode (100 μm) on a standard battery tester (LAND-CT2001A) at different temperatures controlled by cryostat (−50 °C), fridge (−30 °C) and oven (25 °C, 45 °C). The separator used was glass fiber (Waterman GF/D). The cycling stability was evaluated in symmetric cells composed of two identical Zn disks in the electrolyte of TMP−0 and TMP−40 at different current densities and capacities. For the measurement of Coulombic efficiency, the coin cells were composed of titanium (Ti) foil as the working electrode (substrate for Zn plating and stripping), and a piece of Zn foil as the counter and reference electrode. A constant current with a constant capacity (the amount of Zn deposited) was applied to the electrode, followed by Zn stripping via charging to 1 V (versus Zn/Zn$^{2+}$). The Coulombic efficiency of each cycle was calculated as the amount of Zn stripped (based on capacity extracted) divided by the amount of Zn plated (based on capacity deposited) onto the Ti foil. Linear sweep voltammetry (LSV) of the hybrid electrolytes was measured using Ti‖Zn asymmetric at a scan rate of 5 mV s$^{-1}$ in a voltage range of −0.5 to 3 V (versus Zn/Zn$^{2+}$) on an electrochemical workstation (Ivium-n-Stat, Nederlanden).

The ionic conductivity of hybrid electrolytes at different temperatures was measured by EIS via symmetric cells consisting of two parallel Pt-plate electrodes (10 mm × 10 mm). The distance between the two electrodes is 10 mm, and the applied frequency range was from $10^5$ Hz to $10^{-1}$ Hz with 5 mV AC amplitude. The ionic conductivity of the hybrid electrolyte was calculated by the following equation:

$$\sigma = \frac{L}{R_s \times A} \quad (1)$$

where $\sigma$ is the ionic conductivity of the electrolyte (S cm$^{-1}$), $R_s$ is the electrolyte resistance ($\Omega$), which corresponds to the intercept of the Nyquist plot. $L$ is the distance between two Pt-plate electrodes (cm). $A$ stands for the area of Pt electrode (cm$^2$).

The electrochemical performance of Zn−KVOH full cell was evaluated by using both the 2032-type coin cells and pouch cells. For the routine test of the coin-type full cell, the typical mass loading of active materials (KVOH) is around 1.2–1.5 mg cm$^{-2}$, glass fiber (Waterman GF/D) as separator and 100 μm Zn metal as anode. The electrolyte was fixed to around 130 μL. The galvanostatic discharge/charge tests were performed using LAND-CT2001A instruments in a voltage range of 0.2–1.6 V (vs Zn/Zn$^{2+}$). Electrochemical impedance spectroscopy (EIS) measurements were carried out on an electrochemical workstation (Ivium-n-Stat, Nederlanden) from $10^5$ Hz to $10^{-2}$ Hz, and the perturbation amplitude was 5 mV. All the tests at different temperatures were carried out after 2 h of resting.

For the test of coin-type full cell with high-loading cathode (33.75 mg cm$^{-2}$), we used carbon cloth as the current collector, thin glass fiber (Waterman GF/A) as the separator, 50 μm Zn metal as anode, and the lean electrolyte (50 μL) is dropped on the cathode side. The gravimetric energy density ($E_g$, Wh kg$^{-1}$) was calculated according to the following equation:

$$E_g = \frac{VC}{m} \quad (2)$$

where $V$ is the average discharge voltage (0.9 V is assumed), $C$ is the areal capacity (mAh cm$^{-2}$), and $m$ is the areal loading of the active substance (mg cm$^{-2}$).

## Computational details

**Molecular dynamics simulations.** In this work, the solvation environment of TMP−0 (molar ratio, Zn(OTf)$_2$:H$_2$O = 1:27.79) and TMP−40 (molar ratio, Zn(OTf)$_2$:TMP:H$_2$O = 1:1.74:16.67) were simulated. The TMP−0 model system involves 70 Zn(OTf)$_2$ salt molecules and 1945 H$_2$O molecules, while the TMP−40 one contains 70 Zn(OTf)$_2$ salt molecules, 122 TMP molecules, and 1167 H$_2$O molecules. Packmol[52] was used to build the initial configuration of the two model systems. LAMMPS[53] was used to perform the molecular simulations. All-atom molecular dynamics simulations were carried out using PCFF-INTERFACE force field[54]. Constant NVT conditions are enforced using a Nosé-Hoover thermostat with a relaxation time of 100 fs and a temperature of 300 K[55]. The density of each electrolyte solution was determined from 500 ps molecular dynamics simulation in the NPT ensemble at the same thermodynamic conditions. Equations of motions were integrated using the velocity−Verlet method with a 1.0 fs time step. The cut-offs for all the non-bonded interactions are 12 Å. All results reported here are statistical averages taken from runs of 1000 ps in length, each preceded by 1500 ps of equilibration. The atomic coordinates were collected every 1.0 ps for statistical analysis.

**Quantum chemistry calculations.** For the relative binding energy of Zn$^{2+}$ with different species, the structures were fully optimized by using the B3LYP[56,57] method in the level of 6−31 + + G (d, p) basis set. Analytical vibrational frequency was calculated at the same level. The Zn$^{2+}$ cation carries two unit of positive charge. The interaction energy was calculated as follows:

$$\Delta E_{inter} = E_{total} - \left( E_{Zn} + E_{solvent} \right) \quad (3)$$

$\Delta E_{inter}$ represents the interaction energy, $E_{total}$, $E_{Zn}$, and $E_{solvent}$ are the energies of the complex, Zn$^{2+}$ cation, and solvent (H$_2$O or TMP), respectively. The more negative the magnitude of interaction energy, the more favorable the interaction between the Zn$^{2+}$ cation and the solvent is. All calculations were performed with the Gaussian 09 program.

For the migration energy barrier of zinc ions between different components, we have employed the Vienna Ab initio Simulation Package (VASP 6) to perform all density functional theory (DFT) calculations within spin-polarized frame[58,59]. The elemental core and valence electrons were represented by the projector augmented wave (PAW) method and plane−wave basis functions with a cutoff energy of 400 eV. For closed-shell ions (Mg$^{2+}$, Zn$^{2+}$, Al$^{3+}$, Ga$^{3+}$), due to the absence of lone pair electrons, their Ueft values are 0 eV. Therefore, generalized gradient approximation with the Perdew−Burke−Ernzerhof (GGA-PBE) exchange−correlation functional was employed in all the calculations[60,61]. Geometry optimizations were performed with the force convergence smaller than 0.05 eV/Å, where the same convergency was applied for the locating of transition states through the constrained optimizations (NEB). The atoms at the bottom of Zn$_3$(PO$_4$)$_2$ (−121) surfaces (43 atoms) and the ZnF$_2$ (111) surfaces (44 atoms) were fixed in all the calculations. Monkhorst−Pack $k$-points of 3 × 2 × 1 and 2 × 2 × 1 were applied for all the surface calculations for Zn$_3$(PO$_4$)$_2$ (−121), and ZnF$_2$ (111), respectively. To obtain transition states, the bulk structures of Zn$_3$(PO$_4$)$_2$ (−121) and ZnF$_2$ (111) have been optimized with the Monkhorst−Pack $k$-point of 3 × 3 × 3 and 5 × 3 × 8, respectively.

The calculation of the lowest unoccupied molecular orbital (LUMO) in this work was performed using the Gaussian 09 program package. Full geometry optimizations were performed to locate all the stationary points, using B3LYP-D3[62]/6-31 G(d)[63,64]. The Gibbs free energies corrected by zero-point vibrational effects at the M06-2x/6-311 + G(d,p) (D3) level were used in the discussions. The molecular orbital was calculated using the Multiwfn, and VMD software was used for visualization.

## Data availability

All data that support the findings of this study are present in the paper and the Supplementary Information. Additional data related to the study are available from the corresponding author upon reasonable request. Source data are provided with this paper.

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

## Acknowledgements

This work was supported by the Ministry of Science and Technology of China (2021YFA1201900, H.W. and F.W.), National Natural Science Foundation of China (No. 22105107, H.W.), the Science and Technology Commission of Shanghai Municipality (No. 21511103300, F.W.), and Fundamental Research Funds for the Central Universities (H.W.). H.W. acknowledge the Young Elite Scientists Sponsorship Program by Tianjin. The authors thank Professor Jinxiong Wu and Mr. Jiabiao Chen for their help in the operation of the focused ion beam (FIB).

## Author contributions

H.W. and W.W. conceived the original idea. W.W., F.W. and H.W. designed all the experiments. W.W., S.C., X.L.L. and J.L.C. carried out the experiments. W.W., S.C., X.L.L., R.H., F.M.W., Y.X.W., F.W. and H.W. analyzed the experimental data. W.W., F.W. and H.W. co-wrote the paper. All authors were involved in the discussion of the experimental results and preparation of the manuscript.

## Competing interests

The authors have patent filings to disclose. H.W. and W.W. are the inventors on the CN patent application (No. CN115588785 A) regarding the preparation of low-temperature electrolytes for aqueous Zn metal batteries described in this article.
