## [Peer Review File · Nature Communications]

Reviewers' comments:

Reviewer #1 (Remarks to the Author):

The authors introduced trimethyl phosphate additive into the aqueous electrolyte to engineer the solid electrolyte interphase (SEI) for developing Zn-ion battery to operate at a low temperature. The study is interesting and the findings can contribute to develop better Zn-ion batteries. However, a few points need to further address to establish the credibility of the proposed research before it can be considered for publication.

1. The authors used the PCFF-INTERFACE force field without any validation. Use of any off-the-shelf forcefields must be validated for the intended applications. For example, how do the solvation energies and solvation sheath geometries predicted by the PCFF compare with high-level DFT calculations? This is a mandatory requirement.
2. Snapshots of all the images for Figure 4b (NEB) should be provided. Can the authors also provide the geometric coordination of optimized solvation structure and NEB images in the SI?
3. Can the authors justify the use of energy cutoff and k-points for their VASP calculations?
4. The authors only employed fundamental Perdew-Burke-Ernzerhof (PBE) functional which is not suitable for dealing with the localized d-orbitals of Zn. Why not PBE+U? Do the results remain valid if PBE+U is used?

Reviewer #2 (Remarks to the Author):

In this work, the authors introduced trimethyl phosphate (TMP) into the aqueous electrolyte as the co-solvent to modify the solvation structure and form $\text{ZnF}_2\text{-Zn}_3(\text{PO}_4)_2$ SEI in aqueous Zn-ion batteries. However, TMP and similar phosphate solvent have already reported. Furthermore, the electrochemical performance of the resultant electrodes and devices is not superior to previously reported results. Therefore, I cannot recommend its publication at Nature communication. My detailed comments are follows.

- This work by using TMP to tune the solvation structure and SEI of aqueous electrolyte is not new. TMP or similar TEP is already reported in aqueous zinc ion batteries (e.g. 10.1002/adma.201900668, 10.1039/D2SC04143J, 10.1016/j.cej.2022.137843, 10.1002/anie.201813223, 10.1002/adfm.202104281), and even on aqueous lithium ion batteries (e.g. 10.1002/anie.202214126).
- The solvation structure and the formation mechanism of the $\text{ZnF}_2\text{-Zn}_3(\text{PO}_4)_2$ SEI is clear, more clear investigations need.
- Although around 6000 hrs are achieved for Zn metal anode, but the Zn utilization rate is quite low considering the areal capacity of 0.4 mAh cm^{-2} compared to the Zn metal anode (100 μm), which is no meaningful for practical application.
- For the full cell, the KVOH is not a typical cathode for zinc ion battery considering its low work voltage and low capacity. Also the N/P ratio should be given.

Response letter to reviewers

We sincerely thank the reviewers for raising the constructive comments, which have been fully addressed in our revised manuscript. The point-by-point reply to comments is summarized below.

Reviewer#1

Overall comments:

The authors introduced trimethyl phosphate additive into the aqueous electrolyte to engineer the solid electrolyte interphase (SEI) for developing Zn-ion battery to operate at a low temperature. The study is interesting and the findings can contribute to develop better Zn-ion batteries. However, a few points need to further address to establish the credibility of the proposed research before it can be considered for publication.

Response:

We greatly appreciate the reviewer's positive evaluation of our work and kind recommendation. We have performed additional calculations and supplemented detailed discussions to address the reviewer's comments point-by-point.

Comment (1)

The authors used the PCFF-INTERFACE force field without any validation. Use of any off-the-shelf forcefields must be validated for the intended applications. For example, how do the solvation energies and solvation sheath geometries predicted by the PCFF compare with high-level DFT calculations? This is a mandatory requirement.

Response:

We are grateful to the reviewer for the valuable comments. According to the reviewer's advice, we have supplemented additional calculations to confirm the accuracy of the PCFF-INTERFACE force field on the system. The solvation energy of Zn^{2+} in $\text{Zn}(\text{OTf})(\text{TMP})(\text{H}_2\text{O})_4$ cluster and solvation sheath geometries of $\text{Zn}(\text{OTf})(\text{TMP})(\text{H}_2\text{O})_4$ cluster were simulated using high-level DFT and PCFF-INTERFACE force fields, respectively (**Fig. R1**). Obviously, in the prediction of solvation energy, the deviation of the two simulation methods is within 8.4%. In the prediction of solvation sheath geometries, except for a certain deviation (0.4 Å) in the distance between Zn^{2+} and OTf^- , other structural parameters are very consistent. Therefore, considering the approximation of the force field, the molecular dynamics research results using PCFF-INTERFACE force field in our manuscript study are acceptable.

We have included Fig. R1 as Supplementary Fig. S15 and added the corresponding discussion in the revised manuscript (see the Line 24 on Page 7).

39
 40 **Fig. R1.** Optimized geometric configurations (distance in Å) and solvation energies of the
 41 representative Zn(OTf)(TMP)(H₂O)₄ cluster with (a) PCFF-INTERFACE force field and (b)
 42 B3LYP/6-31++G (d, p).

43
 44 **Comment (2)**

45 *Snapshots of all the images for Figure 4b (NEB) should be provided. Can the authors also*
 46 *provide the geometric coordination of optimized solvation structure and NEB images in the SI?*
 47

48 **Response:**

49 We thank the reviewer's valuable comments very much. Based on the suggestion of the
 50 reviewer, the specific migration models for Zn²⁺ in ZnF₂ and Zn₃(PO₄)₂ were provided (**Fig. R2**),
 51 corresponding to the migration potential barriers at different stages in Figure 3D, respectively.

52 We have included Fig. R2 as Supplementary Fig. S26 and added the corresponding discussion
 53 in the revised manuscript (see the Line 23 mark in blue on Page 9).
 54

55
 56 **Fig. R2.** (a) Simulation of Zn²⁺ migration path in ZnF₂ and (b) corresponding migration models
 57 from A to B. (c) Simulation of Zn²⁺ migration path in Zn₃(PO₄)₂ and (d) corresponding migration
 58 models from A to B.

59
 60 **Comment (3)**

61 *Can the authors justify the use of energy cutoff and k-points for their VASP calculations?*
 62

63 **Response:**

64 We very much appreciate the reviewer's valuable comments. We referred to relevant papers
 65 on energy cutoff and k-points values selected for VASP calculations of migration energy barriers
 66 (*Energy Environ. Sci.* 2020, 13, 503–510; *Nat. Commun.* 2020, 11, 3297; *Adv. Mater.* 2021, 33,
 67 2007406; *Adv. Mater.* 2021, 33, 2007416; *Adv. Mater.* 2023, 35, 2207908; *Angew. Chem. Int. Ed.*
 68 2023, 135, e202215324; *Energy Environ. Sci.* 2023, 16, 275–284; *Adv. Funct. Mater.* 2023,
 69 2213416), as displayed in **Table R1**. Meanwhile, we calculated the energy of the model different
 70 energy cuts and k-points for our system, ultimately selecting the appropriate parameters.

71
 72 **Table R1.** Reports related to density functional theory (DFT) calculations of Zn metal batteries

Work	Energy cutoff	Method of calculation
Energy Environ. Sci. 2020, 13, 503–510	400 eV	Perdew-Burke-Ernzerhof (PBE) functional
Nat. Commun. 2020, 11, 3297	400 eV	Perdew-Burke-Ernzerhof (PBE) functional

Adv. Mater. 2021, 33, 2007406	400 eV	Perdew-Burke-Ernzerhof (PBE) functional
Adv. Mater. 2021, 33, 2007416	400 eV	Perdew-Burke-Ernzerhof (PBE) functional
Adv. Mater. 2023, 35, 2207908	400 eV	Perdew-Burke-Ernzerhof (PBE) functional
Angew. Chem. Int. Ed. 2023, 135, e202215324	400 eV	Perdew-Burke-Ernzerhof (PBE) functional
Energy Environ. Sci. 2023, 16, 275–284	400 eV	Perdew-Burke-Ernzerhof (PBE) functional
Adv. Funct. Mater. 2023, 2213416	400 eV	Perdew-Burke-Ernzerhof (PBE) functional

73

74 **Comment (4)**

75 *The authors only employed fundamental Perdew-Burke-Ernzerhof (PBE) functional which is not*
76 *suitable for dealing with the localized d-orbitals of Zn. Why not PBE+U? Do the results remain*
77 *valid if PBE+U is used?*

78

79 **Response:**

80 We are grateful to the reviewer's comments. For closed shell ions (Mg^{2+} , Zn^{2+} , Al^{3+} , Ga^{3+}),
81 due to the absence of lone pair electrons, their Ueff values are 0 eV. Meanwhile, for most
82 reports on DFT calculations related to Zn metal batteries (*Energy Environ. Sci.* 2020, 13, 503–
83 510; *Nat. Commun.* 2020, 11, 3297; *Adv. Mater.* 2021, 33, 2007406; *Adv. Mater.* 2021, 33,
84 2007416; *Adv. Mater.* 2023, 35, 2207908; *Angew. Chem. Int. Ed.* 2023, 135, e202215324;
85 *Energy Environ. Sci.* 2023, 16, 275–284; *Adv. Funct. Mater.* 2023, 2213416), they also only
86 employed fundamental Perdew-Burke-Ernzerhof (PBE) functional (**Table R1**). Therefore, we
87 believe that the fundamental Perdew-Burke-Ernzerhof (PBE) functional is suitable for dealing
88 with the localized d-orbitals of Zn. Supplementary explanations have been provided in the
89 **Methods** regarding theoretical calculations in the revised manuscript (see the Line 1-2 mark in
90 blue on Page 21).

91

92

93 **Reviewer #2**

94 **Overall comments:**

95 *In this work, the authors introduced trimethyl phosphate (TMP) into the aqueous electrolyte as*
96 *the co-solvent to modify the solvation structure and form ZnF_2 - $Zn_3(PO_4)_2$ SEI in aqueous Zn-ion*
97 *batteries. However, TMP and similar phosphate solvent have already reported. Furthermore, the*
98 *electrochemical performance of the resultant electrodes and devices is not superior to previously*
99 *reported results. Therefore, I cannot recommend its publication at Nature communication. My*
100 *detailed comments are follows.*

101

102 **Response:**

103 We greatly appreciate the reviewer's valuable comments. As mentioned by the reviewer, TMP
104 and similar phosphate solvent have been reported for zinc ion batteries. However, most of the
105 works focus on the physical protection of the as-formed phosphate interface to suppress
106 hydrogen evolution reaction and Zn dendrite growth at room temperature. **The interfacial**
107 **behaviors of bivalent Zn^{2+} including Zn^{2+} desolvation and conduction is severely neglected,**
108 **which is very critical for low-temperature Zn batteries.** This is very important for Zn
109 batteries for stationary energy storage in cold climates or high-latitude regions rich in
110 renewable energy. Herein, TMP is only served as a medium to lower the freezing point of water
111 and regular the Zn^{2+} -solvation structure. We care more about **how to optimize the Zn^{2+} kinetics**

112 **across the electrode/electrolyte interface for long-life Zn metal batteries under harsh**
 113 **conditions, rather than simply forming SEI on Zn surface.** To the best of our knowledge,
 114 there are few studies on the SEI engineering for low-temperature ZMBs. The novelty of this
 115 work includes the following three aspects:

116 (1) TMP was selected as a cosolvent to not only decrease the freezing point of aqueous
 117 electrolyte to **-56.8 °C** through breaking H-bonds of water, but also build the solvation structure
 118 of $\text{Zn}^{2+}[\text{H}_2\text{O}]_{5.01}[\text{TMP}]_{0.14}[\text{OTf}^-]_{0.85}$ that enables the **sequential formation of $\text{Zn}_3(\text{PO}_4)_2$ and**
 119 **ZnF_2** due to the difference in oxidative activity of TMP and OTf^- . Moreover, it is for the first
 120 time found that low temperature favors the formation of dense and uniform $\text{ZnF}_2\text{-Zn}_3(\text{PO}_4)_2$ SEI
 121 on Zn metal.

122 (2) The combination of theoretical and experimental studies reveals **the outer ZnF_2 facilitates**
 123 **Zn^{2+} desolvation and the inner $\text{Zn}_3(\text{PO}_4)_2$ servers as channels for fast Zn^{2+} conduction,**
 124 **which was for the first time to be reported with favorable kinetics** for low-temperature
 125 ZMBs cycling.

126 (3) The gradient and dense SEI provides **a record stability in the symmetric Zn cell at -50 °C**
 127 **with a lifespan of 7000 hours (~10 months), and a negligible capacity decay in low-**
 128 **temperature full cells over 12000 cycles, far exceeding those of reported low-temperature**
 129 **aqueous ZMBs (Fig. R3). Moreover, a high Zn utilization rate of 94% at -30 °C was**
 130 **achieved.** Full cells with lean electrolyte and low Zn excess also demonstrate the practical
 131 feasibility.

132
 133 **Fig. R3.** A comparison of our results with the reported literatures for (a) symmetric cells and (b)
 134 full cells. No. 1–16 indicate the data from literatures cited in our manuscript from the references
 135 of 4, 8, 12–17, 19, 22, 42, 43, 45–48.

136 We thank the reviewer's valuable comments and highly valuable suggestions. We have
 137 performed additional experiments and supplemented detailed discussions to address the
 138 reviewer's comments point-by-point.

139

140

141 **Comment (1)**

142 *This work by using TMP to tune the solvation structure and SEI of aqueous electrolyte is not new.*
143 *TMP or similar TEP is already reported in aqueous zinc ion batteries (e.g.*
144 *10.1002/adma.201900668, 10.1039/D2SC04143J, 10.1016/j.cej.2022.137843,*
145 *10.1002/anie.201813223, 10.1002/adfm.202104281), and even on aqueous lithium ion batteries*
146 *(e.g. 10.1002/anie.202214126).*

147

148 **Response:**

149 We thank the reviewer's comments and for bringing these important and pertinent references
150 into our attention. These papers deep understanding on Zn metal batteries, some of which have
151 been cited in our manuscript. Indeed, a few works have reported the use of TMP to tune the
152 solvation structure and SEI of aqueous electrolyte. However, most of these works focus on the
153 result of SEI formation and the role of physical protection to inhibit HER and Zn dendrite growth.
154 The following questions remains to be unsolved: *i) How to harvest the difference in oxidation*
155 **ability of TMP co-solvent and coordinated anions** for a kinetically-favorable SEI towards
156 stable Zn batteries under harsh conditions? *ii) How the SEI compositions and distributions*
157 **affect the Zn²⁺ transport and conduction as well as electrochemical performance?** *iii) Since*
158 *the cycling life of low-temperature Zn metal anode is limited to hundreds of hours, how to*
159 **improve the cycling life of low-temperature Zn metal anode?** This is critically
160 important for Zn metal batteries as the promising option for stationary energy storage in
161 cold climates or high-latitude regions rich in renewable energy.

162 To this end, we optimized the content of TMP co-solvent that cannot break the H-bonds
163 to endow the hybrid electrolyte with a low freezing point of -56.8 °C, but also can regulate the
164 Zn²⁺-solvation structure with a configuration of Zn²⁺[H₂O]_{5.01}[TMP]_{0.14}[OTf⁻]_{0.85}. We find that
165 the lowest unoccupied molecular orbital (LUMO) energy of Zn²⁺-TMP in the solvation shell is
166 lower than that of Zn²⁺-OTf⁻, which can be preferentially reduced on the Zn metal surface,
167 followed by the reductive reaction of OTf⁻, thus forming the gradient inerphase with Zn₃(PO₄)₂
168 in the bottom and ZnF₂ on the top. Moreover, the combination study of experimental
169 characterizations and calculation results reveals that the outer ZnF₂ promotes the desolvation of
170 Zn²⁺ on the interface and inner Zn₃(PO₄)₂ facilitates rapid transport across the SEI, respectively.
171 Consequently, it achieves an average Coulombic efficiency of 99.9% over 3800 cycles and a
172 high Zn utilization rate of 94% at -30 °C, and remarkable durability over 7000 hours at -50 °C,
173 which represent the best low-temperature ZMBs performance to the best of our knowledge.
174 High-capacity full cells with KVOH as cathode were also demonstrated with superb capacity
175 retention ability. We also compared the electrochemical performance of symmetric Zn cell
176 and full cell at wide temperatures with those recommended works, as shown in **Table R2.**
177 **It can be clearly seen that the devices are superior to previously reported results.**

178 Therefore, we believe that our work offers new insights on how to regulate interfacial reaction
179 through electrolyte chemistry for stable and low-temperature Zn metal batteries towards practical
180 applications. The new findings here would be instructive of broad interest to the field of
181 rechargeable batteries especially working under harsh conditions.

182 **Table R2.** Comparison of the electrochemical performance in this work with those
183 recommended papers at wide temperatures

Work	Electrolyte	Temperature	Coulombic efficiency	Cyclic stability	Full cells
1	0.5 M	25 °C	99.5%	2000 h @1 mA cm ⁻² /~ mAh cm ⁻²	500 cycle@0.1 A g ⁻¹

	Zn(OTf) ₂ - TMP		500 cycles		
2	0.5 M Zn(ClO ₄) ₂ ·6H ₂ O-TMP	25-50 °C	99.5% 500 cycles (25 °C)	3000 h @1 mA cm ⁻² /1 mAh cm ⁻² (25 °C); 3000 h @5 mA cm ⁻² /5 mAh cm ⁻² (50 °C)	1000 cycles@1 A g ⁻¹ (25 °C) 50 cycles@0.5 A g ⁻¹ (50 °C)
3	0.5 M Zn(CF ₃ SO ₃) ₂ - TEP	25 °C	99.68% 1000 cycles	2000 h@0.5 mA cm ⁻² / ~ 0.25mAh cm ⁻²	1000 cycles@1 C
4	0.5 M Zn(OTf) ₂ - TEP+H ₂ O=1: 1	25 °C	99.5% 200 cycles	1500 h @1 mA cm ⁻² /1 mAh cm ⁻²	1000 cycles@5 A g ⁻¹
5	2 M Zn(OTf) ₂ - TMP+H ₂ O=1 :1	0-25 °C	99.57% 200 cycles (25 °C)	300 h @10 mA cm ⁻² /10 mAh cm ⁻² (25 °C)	8000 cycles@10 A g ⁻¹ (25 °C) 1200 cycles@1 A g ⁻¹ (0 °C)
6	9.5 M LiTFSI-TMP- H ₂ O	25 °C	N/A	N/A	1000 cycles@5 C
Our work	2 M Zn(OTf)₂- TMP+H₂O= 4:6	-50-45 °C	99% 600 cycles (45 °C); 99.5% 1000 cycles (25 °C); 99.9% 3800 cycle (-30 °C); 99% ~500 cycle (-50 °C)	450 h @5 mA cm⁻²/1 mAh cm⁻² (45 °C); 500 h @5 mA cm⁻²/1 mAh cm⁻² (25 °C); 3600 h @2 mA cm⁻²/2 mAh cm⁻² (-30 °C); 7000 h @0.4 mA cm⁻²/0.4 mAh cm⁻² (-50 °C)	800 cycles@10 A g⁻¹ (45 °C) 2300 cycles@1 A g⁻¹ (25 °C) 10800 cycles@10 A g⁻¹ (-30 °C) 12000 cycles@1 A g⁻¹ (-50 °C)

184 Note: work 1–6 is from doi: 10.1002/adma.201900668; 10.1039/D2SC04143J;
 185 10.1002/anie.201813223; 10.1002/adfm.202104281; 10.1016/j.cej.2022.137843;
 186 10.1002/anie.202214126, respectively.

187

188 **Comment (2)**

189 *The solvation structure and the formation mechanism of the ZnF₂-Zn₃(PO₄)₂ SEI is clear, more*
 190 *clear investigations need.*

191

192 **Response:**

193 We are grateful to the reviewer for the worthy comments. As suggested by the reviewer, we
 194 conducted DFT calculation to compare the lowest unoccupied molecular orbital (LUMO) energy
 195 level of the components in Zn²⁺ solvated sheath. As displayed in **Fig. R4**, the LUMO energy
 196 level of Zn²⁺-TMP is much lower than that of Zn²⁺-OTf⁻. Meanwhile, the coordinated TMP in
 197 the Zn²⁺-solvation sheath is farther from the Zn²⁺ than OTf⁻ (Fig. 2c and **Fig.R5**). **Collectively,**
 198 **the TMP can preferentially accept electrons from the Zn metal to be reduced into**
 199 **Zn₃(PO₄)₂, followed by the decomposition of OTf⁻ into ZnF₂, forming gradient SEI with**
 200 **Zn₃(PO₄)₂ at the bottom and ZnF₂ on the top.** Notably, the LUMO energy of Zn²⁺-OTf⁻ is
 201 lower than free OTf⁻, indicating that the introduction of TMP increases the solvated OTf⁻ to
 202 promote its decomposition into ZnF₂ in aqueous electrolytes. **Fig. R5** shows the schematic of the
 203 solvation structure and the formation mechanism of the ZnF₂-Zn₃(PO₄)₂.

204 We have included Fig. R4 and Fig. R5 as Supplementary Fig. S24, Fig.2e and added the
 205 related discussions in the Line 1-8 on Page 9 in the revised manuscript.

206

207
 208 **Fig. R4.** LUMO energy levels with corresponding isosurfaces of free OTf⁻, free TMP, free H₂O,
 209 Zn²⁺-OTf⁻ and Zn²⁺-TMP coordination.
 210

211
 212 **Fig. R5.** Schematic of the solvation structure and the formation mechanism of the ZnF₂-
 213 Zn₃(PO₄)₂ interlayer.
 214

215 **Comment (3)**

216 *Although around 6000 hrs are achieved for Zn metal anode, but the Zn utilization rate is quite*
 217 *low considering the areal capacity of 0.4 mAh cm⁻² compared to the Zn metal anode (100 μm),*
 218 *which is no meaningful for practical application.*
 219

220 **Response:**

221 We greatly appreciate the reviewer's valuable comments. We strongly agree with the
 222 reviewer's opinion that the utilization rate of Zn is crucial for practical applications. We further
 223 conducted the stability test of Zn deposition and stripping under actual conditions with 10 μm of
 224 thickness Zn foil (~5.85 mAh cm⁻²) at -30 °C. As displayed in **Fig. R6**, with a capacity of 4
 225 mAh cm⁻² corresponding to the Zn utilization rate of 68%, the symmetrical cell with TMP-40
 226 electrolyte exhibited a highly stable voltage profile over 1800 hours. **As Zn utilization rate was**
 227 **increased to 85% and 94%, the cells can still maintain high stability over 500 hours with 5**

228 **mAh cm⁻² and over 180 hours with 5.5 mAh cm⁻².** These results demonstrate the viability of
229 the gradient SEI in stabilizing the ZMBs under actual and harsh conditions.

230 We have included Fig. R6 as Fig. 4g and Supplementary Fig. S38, and added the discussions
231 in the Line 9-16 on Page 12 in the revised manuscript.

232 **Fig. R6.** Galvanostatic cycling stability of symmetrical Zn cells with TMP-40 electrolyte under
233 a utilization of a) 68%, b) 85% and c) 94% at -30 °C.

234
235
236 **Comment (4)**

237 *For the full cell, the KVOH is not a typical cathode for zinc ion battery considering its low work*
238 *voltage and low capacity. Also the N/P ratio should be given.*

239
240 **Response:**

241 We are grateful to the reviewer's comments. **Actually, the N/P ratio for the Zn-KVOH**
242 **cathode has been provided in the manuscript with the description:** In view of the inspiring
243 performance, we further evaluated the application of TMP-40 electrolyte in practical situation by
244 controlling lean electrolyte and low Zn excess. As shown in Fig. 5b, when the KVOH loading
245 increases to 33.75 mg cm⁻², the cell still delivers a superhigh initial areal capacity of 9.42 mAh
246 cm⁻¹ with lean E/C (6.76 μL mAh⁻¹, the ratio of electrolyte volume to capacity) ratio and low
247 N/P (3.1, the ratio of negative to positive). The corresponding energy density is calculated to be
248 251.2 Wh kg⁻¹ (based on the KVOH mass) with a high capacity retention of 93.3% after 50
249 cycles. The outstanding performance can be ascribed to the superior kinetics and great robustness
250 of the ZnF₂-Zn₃(PO₄)₂ SEI that can allow large amounts of Zn²⁺ to repeatedly strip and plate.

251 Inspired by the reviewer's comment, we further extended the TMP-40 electrolyte to the
252 commonly used MnO₂ cathode with a high voltage. As shown in **Fig. R7**, the Zn-MnO₂ full cell
253 still maintains an areal capacity of 3.41 mAh cm⁻² after 40 cycles with a high MnO₂ loading
254 (20.4 mg cm⁻²) and low N/P (3.2). Notably, the paired anode is 20 μm-thickness Zn foil that
255 corresponds to an areal capacity of 11.7 mAh cm⁻². These results fully demonstrate the **TMP-40**
256 **electrolyte is promising for the practical application.**

257 We have included Fig. R7 as Supplementary Fig. S47 and added the related discussions in the
258 Line 14-16 on Page 14 in the revised manuscript.

259
 260 **Fig. R7.** Zn-MnO₂ full cell test under practical conditions with TMP-40 electrolyte. (a) Cycling
 261 performance. (b) Charge-discharge curves with different cycles.

REVIEWER COMMENTS

Reviewer #1 (Remarks to the Author):

The authors addressed my previous comments adequately.

Reviewer #3 (Remarks to the Author):

In this study, the authors introduced a novel approach to modify the components of the SEI and enhancing the kinetic performance of Zinc electrodes in harsh conditions by controlling the order of interfacial chemical reactions. This work expands the application of Zn-ion batteries in extreme environments and improves the utilization of the Zn anode, which helps to build better Zn-ion batteries.

I believe it can become suitable for Nature Comm after the following issues are addressed.

1. The author compared the Zn²⁺ desolvation energy of TMP-0 and TMP-40 by extracting the respective R_{ct} before cycling, please cite literature to show the source of this method. Also, the equivalent circuit model in Supplementary Figure 16. c should contain the electrolyte resistance R₀, like the one in Supplementary Figure 25. c.
2. SEI was formed at the initial plating (Line 193, page 8). The LSV curves of Ti | Zn asymmetric cells were performed before and after 5th cycling in the TMP-40 electrolyte. How about the second and the third scan curve? Does the SEI form only in the first cycle? Also, the scan rate should be provided.
3. The author compared the kinetic behavior of bivalent Zn²⁺ on the electrode/electrolyte interface using charge transfer resistance (R_{ct}) and the resistance associated with Zn²⁺ crossing SEI (R_{SEI}). Since there is no SEI formed for the reference Zinc electrode as the author indicated, what's the meaning of R_{SEI} for the reference sample TMP-0 (Supplementary Figure 25)?
4. The high-resolution transmission electron microscopy (HR-TEM) of the Zn electrode surface don't support the gradient ZnF₂-Zn₃(PO₄)₂ interphase clearly. A clear interface of Zinc and SEI layers should be provided, like previous research on the Zinc-SEI modification (doi.org/10.1038/s41565-021-00905-4). Especially, the EDS mapping in Supplementary Figure 21. c can't help to support the gradient ZnF₂-

Zn₃(PO₄)₂ interphase. An EELS mapping or line scan from the electrode surface to the bulk Zinc should be provided. It is suggested to prepare a lamella sample for HR-TEM testing, which may help to recognize the structure of gradient ZnF₂–Zn₃(PO₄)₂ interphase. In addition, the HR-TEM tests for the reference TMP–0 should be provided for comparison.

5. Some misstatements should be corrected like “The TMP co-solvent cannot break the H-bonds to endow the hybrid electrolyte with a low freezing point. (Line 85, page 4)” Also, the authors marked the crystal zone axis in Supplementary Figure 21b (viewed along the [110] direction). This is not an exact practice, in the case of a small nanocrystal, we cannot accurately judge the orientation of its crystal zone axis, especially in the case of mixing with other nanocrystals (ZnF₂ and Zn₃(PO₄)₂).

Response letter to reviewers

We sincerely thank the reviewers for raising the constructive comments, which have been well addressed in our revised manuscript. The point-by-point reply to comments is summarized below.

Reviewer #1

Overall comments:

The authors addressed my previous comments adequately.

Response:

We are grateful to the reviewer's valuable comments that have helped to greatly improve our manuscript.

Reviewer #3

Overall comments:

In this study, the authors introduced a novel approach to modify the components of the SEI and enhancing the kinetic performance of Zinc electrodes in harsh conditions by controlling the order of interfacial chemical reactions. This work expands the application of Zn-ion batteries in extreme environments and improves the utilization of the Zn anode, which helps to build better Zn-ion batteries. I believe it can become suitable for Nature Comm after the following issues are addressed.

Response:

We greatly appreciate the reviewer's positive evaluation of our work and highly valuable suggestions. We have performed additional experiments and supplemented detailed discussions to address the reviewer's comments point-by-point.

Comment (1)

The author compared the Zn^{2+} desolvation energy of TMP-0 and TMP-40 by extracting the respective R_{ct} before cycling, please cite literature to show the source of this method. Also, the equivalent circuit model in Supplementary Figure 16. c should contain the electrolyte resistance R_0 , like the one in Supplementary Figure 25. c.

Response:

We greatly appreciate the reviewer's valuable comments. According to the reviewer's suggestion, we cited the relevant papers (*J. Electrochem. Soc.* 2004, 151, A1120–A1123; *J. Am. Chem. Soc.* 2019, 141, 9422–9429) as References 36 and 37 in the revised manuscript. We are very sorry to use a different annotation in the equivalent circuit model of *Supplementary Fig. 16c* in the original manuscript, where the R_1 should be R_0 . We have modified the equivalent circuit model as shown in Figure R1c. We have included Fig. R1 as Supplementary Fig. S16 and added the corresponding explanations in the caption (see Page 17 in the revised Supporting Information highlighted in yellow).

The fitting results corresponding to R_{ct} are as follows:

Temperature (K)		243	253	263	273	283	293
TMP-0	R_{ct} (Ω)	299900	66934	25552	9941	4477	1408
TMP-40	R_{ct} (Ω)	827930	302790	53342	22825	9720	2837

Figure R1. Comparison of desolvation energy between TMP-0 and TMP-40 electrolytes. Temperature-dependent electrochemical impedance spectra of Zn||Zn symmetric cells with (a) TMP-0 or (b) TMP-40 electrolyte. (c) The equivalent circuit model. R_0 represents bulk resistance of the cell, which reflects electric conductivity of the electrolyte, separator and electrodes; R_{ct} and C_{dl} stands for faradic charge-transfer resistance and its relative double-layer capacitance, respectively. (d) Arrhenius fitting of R_{ct} derived from the Nyquist plots of the Zn||Zn symmetric cells with TMP-0 and TMP-40 electrolytes.

Comment (2)

SEI was formed at the initial plating (Line 193, page 8). The LSV curves of Ti||Zn asymmetric cells were performed before and after 5th cycling in the TMP-40 electrolyte. How about the second and the third scan curve? Does the SEI form only in the first cycle? Also, the scan rate should be provided.

Response:

We very much appreciate the reviewer's careful reading. To explore the SEI formation process, we further conducted LSV tests on Ti||Zn asymmetric cells after different cycles with TMP-40 electrolyte at a scan rate of 5 mV s^{-1} . As shown in Figure R2a, the peak current density after one cycle of activation reaches the maximum value of 0.334 mA cm^{-2} and gradually decreases as the cycling proceeds. At 5 cycles, the peak current density almost disappears and remains almost unchanged even after 20 cycles (Figure

R2b), representing that the SEI formation reaches the steady state. Also, an obviously negative shift is observed for the onset potential of hydrogen evolution reaction (HER) after 5 cycles compared to that before cycling, indicating the as-formed SEI can effectively suppress HER. These results indicate the SEI formation is mainly formed in the first three cycles and remain stable after 5 cycles.

We have included Fig. R2 as Supplementary Fig. S19 and added the related discussions in the revised manuscript (see Line 7 on Page 8 and Methods Line 7 on Page 19 highlighted in yellow). Meanwhile, additional explanations have also been added in the supporting information (see the highlighted part on Page 20).

Figure R2. (a) LSV curves of Ti||Zn asymmetric cells after 5 cycles with TMP-40 electrolyte at a scan rate of 5 mV s⁻¹. (b) The comparison of LSV curves after 5, 10 and 20 cycles.

Comment (3)

The author compared the kinetic behavior of bivalent Zn²⁺ on the electrode/electrolyte interface using charge transfer resistance (R_{ct}) and the resistance associated with Zn²⁺ crossing SEI (R_{SEI}). Since there is no SEI formed for the reference Zinc electrode as the author indicated, what's the meaning of R_{SEI} for the reference sample TMP-0 (Supplementary Figure 25)?

Response:

We are grateful to the reviewer's comments. As displayed Figure R3, zinc triflate hydroxide hydrate ($Zn_xOTf_y(OH)_{2x-y} \cdot nH_2O$, ZOTH) was detected on the Zn surface in TMP-0. That is, there is passivation film formed on the Zn electrode in TMP-0 electrolyte, as described in the manuscript (please see Line 10 on Page 8 highlighted in yellow). Therefore, R_{SEI} in the reference electrolyte (TMP-0) represents the impedance that Zn²⁺ crosses the ZOTH passivation film. We have included Figure R3 as Supplementary Fig. S20 in the revised manuscript (see Page 21 in the revised Supporting Information highlighted in yellow).

Figure R3. XRD patterns of Zn anodes in TMP-0 and TMP-40 electrolytes after 40 cycles, where the characteristic peaks of ZOTH are observed.

Comment (4)

The high-resolution transmission electron microscopy (HR-TEM) of the Zn electrode surface don't support the gradient $\text{ZnF}_2\text{-Zn}_3(\text{PO}_4)_2$ interphase clearly. A clear interface of Zinc and SEI layers should be provided, like previous research on the Zinc-SEI modification (doi.org/10.1038/s41565-021-00905-4). Especially, the EDS mapping in Supplementary Figure 21. c can't help to support the gradient $\text{ZnF}_2\text{-Zn}_3(\text{PO}_4)_2$ interphase. An EELS mapping or line scan from the electrode surface to the bulk Zinc should be provided. It is suggested to prepare a lamella sample for HR-TEM testing, which may help to recognize the structure of gradient $\text{ZnF}_2\text{-Zn}_3(\text{PO}_4)_2$ interphase. In addition, the HR-TEM tests for the reference TMP-0 should be provided for comparison.

Response:

We greatly appreciate the constructive comments provided by the reviewer and thanks for bringing these important and pertinent references into our attention. We agree with the reviewer's opinion that lamella sample for elemental mappings or line scan from the SEI to the bulk Zn surface as well as the corresponding HR-TEM testing are very essential to recognize the structure of gradient $\text{ZnF}_2\text{-Zn}_3(\text{PO}_4)_2$ interphase. We referred to the relevant papers (*ACS Energy Lett.* 2021, **6**, 3063–3071; *Angew. Chem. Int. Ed.* 2023, **62**, e202215600; *Angew. Chem. Int. Ed.* 2023, e202308017) and found the detailed information of the cross-sectional SEI are all obtained by SEM equipped with focused ion beam (FIB-SEM), TEM, HRTEM, EDS mappings. Therefore, we prepared the lamella sample through FIB-SEM, and further characterized the structures and distributions of the as-formed SEI with TEM, EDS mappings, line scan as well as HRTEM images. As displayed in Figure R4a, FIB cutting technique was employed on the Zn electrode cycled in the TMP-40 electrolyte after 20 cycles at a current density of 1 mA cm^{-2} for cross-sectional analysis. The TEM image in Figure R4b shows the thickness of the SEI layer is around 500 nm. Accordingly, the EDS mappings (Figure R4c) in the selected region reveal that the SEI layer mainly consists of Zn, F, and P elements. Notably, the content of F element predominately distributes in the upper region of the as-formed SEI, while the P element mainly concentrates on the near region of Zn electrode. The HRTEM images of the selected regions of A₁, A₂ and B₁, B₂

derived from Figure R4b identify the lattice fringes of ZnF_2 and $\text{Zn}_3(\text{PO}_4)_2$ (Figure R4d, e), further confirming that ZnF_2 and $\text{Zn}_3(\text{PO}_4)_2$ dominates the top and bottom of the SEI, respectively. Moreover, EDS line scan along the direction of the arrow in Figure R4b shows the F content gradually decreases, while the P content gradually increases as the detection depth of SEI increases (Figure R4f), according well with the with EDS mappings. For comparison, identical operations and characterizations were also applied on the Zn electrode after cycling in TMP–0 electrolyte. Obviously, no obvious F and P signals were observed in EDS mapping and line scan (Figure R5b–d). These results manifest the structure of gradient ZnF_2 – $\text{Zn}_3(\text{PO}_4)_2$ interphase on Zn electrode cycled in the TMP–40 electrolyte, which is consistent with the XPS results with Ar ion sputtering (Figure 2d).

We have included Figures R4 and R5 as Supplementary Figs. S24 and S25, respectively, and added the related discussions in the revised manuscript (see Page 8 and Methods on Page 18 highlighted in yellow). Additional explanations have also been added in the supporting information on Page 25.

Figure R4. (a) SEM image of Zn electrode cycled in the TMP–40 electrolyte after FIB cutting. (b) TEM image and (c) EDS mappings of the SEI. The mapping area was indicated by a red rectangular box. The Pt deposited on the electrode surface is used to protect the SEI from ion beam damage during sample preparation. (d, e) HRTEM images of the selected regions A₁, A₂ and B₁, B₂ derived from Figure R4b. (f) EDS line scan of P and F elements along the direction of white arrow in Figure R4b.

Figure R5. (a) SEM image of Zn electrode cycled in the TMP-0 electrolyte after FIB cutting for TEM analysis. (b) TEM image and (c) EDS mapping of the selected area indicated by a red rectangular box. The Pt deposited on the electrode surface is used to protect the SEI from ion beam damage during sample preparation. (d) Results of EDS line scan along the direction of white arrow in Figure R5b.

Comment (5)

Some misstatements should be corrected like “The TMP co-solvent cannot break the H-bonds to endow the hybrid electrolyte with a low freezing point. (Line 85, page 4)” Also, the authors marked the crystal zone axis in Supplementary Figure 21b (viewed along the [110] direction). This is not an exact practice, in the case of a small nanocrystal, we cannot accurately judge the orientation of its crystal zone axis, especially in the case of mixing with other nanocrystals (ZnF_2 and $Zn_3(PO_4)_2$).

Response:

We are very grateful to the reviewers for pointing out the errors. As suggested by the reviewer, we have made modifications on the descriptions in the revised manuscript (see Line 25 on Page 4 highlighted in yellow) and double checked through the manuscript to avoid the misstatements. Meanwhile, the annotation in Supplementary Fig. 21b has been corrected, where the crystal zone axis is removed, as shown in Fig. R6. We have included it as Supplementary Fig. S21 in the revised manuscript (see Page 22 in the revised Supporting Information highlighted in yellow).

Figure R6. (a) HRTEM image of Zn surface after cycling by Zn||Zn symmetric cells with TMP-40 electrolyte. Inset: the corresponding SAED pattern. (b) The crystal plane spacing of corresponding ZnF_2 and $Zn_3(PO_4)_2$. (c) Element distribution of SEI formed on the Zn surface in TMP-40 electrolyte.

REVIEWERS' COMMENTS

Reviewer #3 (Remarks to the Author):

The authors have answered the questions I raised in my previous review. I feel the manuscript can now be published in Nature Comm.

Response letter to reviewers

Reviewer #3

Overall comments:

The authors have answered the questions I raised in my previous review. I feel the manuscript can now be published in Nature Comm.

Response:

We very much appreciate the reviewer's valuable comments and kind recommendation of our work.